# THE BEST OF BOTH WORLDS: AMORTIZED VARIATIONAL DIFFUSION POSTERIOR SAMPLING

## ABSTRACT

Diffusion models pre-trained on large datasets are powerful priors for inverse problems such as super-resolution and inpainting. Zero-shot *variational diffusion posterior sampling* achieves state-of-the-art reconstructions without task-specific training but is slow due to costly test-time optimization. Supervised diffusion for inverse problems offers fast inference, yet demands large datasets and often fails under unseen degradations. We introduce a *best-of-two-worlds* strategy that jointly leverages upstream training and test-time likelihood guidance. An amortized inference model, trained on a small paired dataset, predicts a good initialization for a variational approximation problem involved in variational diffusion posterior sampling, while retaining the explicit use of the degradation operator to guide inference. This combination removes several gradient updates at inference, yielding up to a $\times 1.31$ speedup compared to zero-shot posterior sampling. Importantly, it remains robust against out-of-distribution degradation operators and training settings with limited data (e.g., $1\%$ of the pre-training data), outperforming the supervised diffusion baselines in these scenarios. Our results show that coupling modest training with test-time operator knowledge can unlock fast, flexible, and high-quality diffusion reconstructions.

## 1 INTRODUCTION

An inverse problem aims to recover an unknown signal from incomplete and noisy observations. Such problems arise in a wide range of applications including, but not limited to, imaging tasks such as super-resolution, inpainting, and deblurring. These problems are typically difficult because they are ill-posed: a single perturbed observation may correspond to multiple plausible reconstructions. To overcome this difficulty, recent work has employed diffusion models (Ho et al., 2020; Nichol & Dhariwal, 2021; Rombach et al., 2022) pre-trained on large-scale image datasets. These models serve as powerful *priors* that guide the solution of inverse problems towards reconstructions that are both realistic and consistent with the observed data.

A prominent line of work is *zero-shot diffusion posterior sampling* (Chung et al., 2023; Kadkhodaie & Simoncelli, 2021; Kawar et al., 2022; Lugmayr et al., 2022; Song et al., 2023; 2021b; Wang et al., 2023), a family of methods that require *no additional training* beyond a pre-trained diffusion model. At test time, these approaches aim to sample from the *posterior* distribution of reconstructions conditioned on the degraded observation. Specifically, diffusion posterior sampling approximates the posterior by progressively transforming noise into data samples, following the same iterative denoising principle as standard diffusion models.

In this framework, the conditional denoiser of the backward diffusion process is decomposed into two components: (i) the prior denoiser provided by the pre-trained model, and (ii) a correction term corresponding to the gradient of the log-likelihood of the observation given the current noisy sample. Because this likelihood term is generally intractable, it is typically approximated through heuristics proposed in prior work (Chung et al., 2023; Rozet et al., 2024; Song et al., 2023). This allows posterior sampling approaches to produce high-quality reconstructions for inverse problems, with the notable advantage of being applicable in a zero-shot manner to arbitrary degradation operators at test time. Within this family of methods, *variational diffusion posterior sampling* (Janati et al., 2025a; Moufad et al., 2025) has emerged as one of the most effective approaches, achieving state-of-the-art reconstruction quality among posterior sampling methods. In contrast to simpler strategies

such as the DPS approximation (Chung et al., 2023)—which models the denoising distribution as a Dirac mass centered at the pre-trained denoiser and can lead to large errors in highly multimodal posteriors—these methods rely on a more accurate approximation of the likelihood (or equivalently, the guidance). However, this improved modeling comes at the cost of a more complex backward process, as it requires the sampling of transition kernels that incorporate the pre-trained denoiser. The procedure is performed by a Gaussian variational approximation, but solving the associated minimization problem with a first-order optimization usually requires several gradient steps through the denoiser, which makes the inference significantly slower than posterior sampling algorithms based on simpler approximations.

**Our contributions.** We perform *amortized optimization* (Amos, 2023) to accelerate variational diffusion posterior sampling. Our approach trains an inference model on a paired dataset and uses it at test time to warm-start the variational approximation, reducing or even eliminating the need for costly gradient-based updates during inference. This design makes three important contributions:

1. **Fast inference for variational posterior sampling.** Our method achieves up to $\times 1.31$ inference time speedup compared to zero-shot variational diffusion posterior sampling, while maintaining the quality of the reconstruction, on a variety of tasks such as super-resolution, inpainting and motion deblurring. This demonstrates the effectiveness of using upstream training data to significantly speed up posterior sampling at test time.

2. **Strong performance in data-scarce regimes.** We show that the amortized inference model can be trained with only $1\%$ of the data used to pre-train the unconditional diffusion model. Remarkably, our method outperforms the fine-tuned supervised diffusion models (Saharia et al., 2022a) in this data-poor regime. To the best of our knowledge, this is the first systematic comparison between posterior sampling and supervised diffusion in a data-poor setting. Our experiments highlight the limitations of fine-tuning in the presence of data scarcity and demonstrate that amortized optimization in combination with posterior sampling is a principled and effective alternative.

3. **Robustness to out-of-distribution degradation operators.** We further reveal a key weakness of supervised diffusion models: even when the degradation operator is explicitly incorporated into the denoiser architecture, as proposed in (Elata et al., 2025), they can collapse when faced with some specific degradations unseen during training. In contrast, our method incorporating the amortized inference model remains robust to such out-of-distribution (OOD) operators, as demonstrated on our inpainting experiments. This highlights how inheriting the test-time flexibility of posterior sampling allows our method to accommodate unseen degradations without retraining and to maintain high reconstruction quality well beyond the training distribution.

## 2 BACKGROUND

**Notations.** The Gaussian distribution of mean $\boldsymbol{\mu}$ and covariance $\boldsymbol{\Sigma}$ is denoted by $\mathcal{N}(\boldsymbol{\mu}, \boldsymbol{\Sigma})$. Its density is denoted as $\boldsymbol{x} \mapsto \mathrm{N}\left(\boldsymbol{x}; \boldsymbol{\mu}, \boldsymbol{\Sigma}\right)$. The identity matrix of size $d$ is denoted as $\boldsymbol{I}_d$. The notation $\boldsymbol{X} \sim p$ indicates that $\boldsymbol{X}$ is a random variable whose probability distribution has density $p$. We use the notation $\propto$ to define a probability density up to a normalization constant.

**Diffusion models.** Denoising Diffusion Models (DDMs) (Ho et al., 2020; Sohl-Dickstein et al., 2015; Song & Ermon, 2019) define a generative process for a data distribution $p_0 := p_{\mathrm{data}}$, initialized from a non-informative base distribution $p_T := \mathcal{N}(0, \boldsymbol{I}_d)$. The model then sequentially samples from increasingly structured intermediate distributions $p_t$ in reverse order until $p_0$ is recovered. The probability path $(p_t)_{t=0}^T$ is determined by the *forward* diffusion process: $p_t(\boldsymbol{x}_t) := \int q_{t|0}(\boldsymbol{x}_t|\boldsymbol{x}_0) p_0(\boldsymbol{x}_0) \, \mathrm{d}\boldsymbol{x}_0$ where we define for all $s < t$,

$$q_{t|s}(\boldsymbol{x}_t|\boldsymbol{x}_s) := \mathrm{N}\left(\boldsymbol{x}_t; \alpha_{t|s}\boldsymbol{x}_s, \sigma_{t|s}^2 \boldsymbol{I}_d\right), \tag{1}$$

with $\alpha_{t|s} := \alpha_t / \alpha_s$, $\sigma_{t|s}^2 := \sigma_t^2 - \alpha_{t|s}^2 \sigma_s^2$ where $(\alpha_s)_{s=0}^T$, $\alpha_0 = 1$ and $(\sigma_t)_{t=0}^T$, $\sigma_0 = 0$ are positive sequences chosen so that the signal-to-noise ratio $(\alpha_t^2/\sigma_t^2)_{t>1}$ decreases monotonically. A popular choice of schedule is $\alpha_t = 1 - t/T$ and $\sigma_t = t/T$ (Esser et al., 2024; Lipman et al., 2023). For $s < t$, define the Gaussian transition $q_{s|0,t}(\boldsymbol{x}_s|\boldsymbol{x}_0, \boldsymbol{x}_t) \propto q_{s|0}(\boldsymbol{x}_s|\boldsymbol{x}_0)q_{t|s}(\boldsymbol{x}_t|\boldsymbol{x}_s)$ and the reverse transitions

$$p_{s|t}(\boldsymbol{x}_s|\boldsymbol{x}_t) := \int q_{s|0,t}(\boldsymbol{x}_s|\boldsymbol{x}_0, \boldsymbol{x}_t) p_{0|t}(\boldsymbol{x}_0|\boldsymbol{x}_t) \, \mathrm{d}\boldsymbol{x}_0, \quad 0 < s < t,$$

which can be shown to satisfy the recursion $p_s(\boldsymbol{x}_s) = \int p_{s|t}(\boldsymbol{x}_s|\boldsymbol{x}_t)p_t(\boldsymbol{x}_t)\,\mathrm{d}\boldsymbol{x}_t$. Given a sample $\boldsymbol{X}_t \sim p_t$, drawing $\boldsymbol{X}_s \sim p_{s|t}(\cdot|\boldsymbol{X}_t)$ yields a sample from the more structured distribution $p_s$. However, these backward transition kernels are intractable due to the integration over the posterior $p_{0|t}(\cdot|\boldsymbol{x}_t)$. Following Ho et al. (2020), the latter is approximated with a Dirac mass at $D_{0|t}^\theta(\boldsymbol{x}_t)$, the neural network approximation of its expectation, *i.e.*, the denoiser $\mathbb{E}[\boldsymbol{X}_0|\boldsymbol{X}_t = \boldsymbol{x}_t]$ where $\boldsymbol{X}_0 \sim p_0$ and $\boldsymbol{X}_t \sim q_{t|0}(\cdot|\boldsymbol{X}_0)$. This yields the transition approximation

$$p_{s|t}^\theta(\boldsymbol{x}_s|\boldsymbol{x}_t) := q_{s|0,t}(\boldsymbol{x}_s|D_{0|t}^\theta(\boldsymbol{x}_t), \boldsymbol{x}_t). \tag{2}$$

The parameters $\theta$ of the denoiser are typically learned by minimizing a weighted $\ell_2$ denoising loss over several time steps:

$$\min_\theta \sum_{t=1}^T w_t\, \mathbb{E}\big[\|D_{0|t}^\theta(\boldsymbol{X}_t) - \boldsymbol{X}_0\|_2^2\big], \tag{3}$$

where $(w_t)_{t=1}^T$ denotes a sequence of non-negative weights. At inference time, the generative process starts with $\hat{\boldsymbol{X}}_T \sim \mathcal{N}(0, \boldsymbol{I}_d)$ and iteratively samples $\hat{\boldsymbol{X}}_{t-1} \sim p_{t-1|t}(\cdot|\hat{\boldsymbol{X}}_t)$ for $t > 1$. Finally, $\hat{\boldsymbol{X}}_0 := D_{0|1}^\theta(\hat{\boldsymbol{X}}_1)$ is used as an approximate sample from $p_0$.

**Bayesian inverse problems.** In the Bayesian formulation of *inverse problems*, the following observation model is considered:

$$\boldsymbol{Y} = \mathcal{A}(\boldsymbol{X}_0) + \sigma_{\boldsymbol{y}}\boldsymbol{W}, \quad \boldsymbol{X}_0 \sim p_0, \quad \boldsymbol{W} \sim \mathcal{N}(0, \boldsymbol{I}_{d'}), \tag{4}$$

where $\mathcal{A}$ is a known *degradation operator* (also called *forward model*) acting on the space of the signal to recover (e.g., the pixel space for image inverse problems), $\sigma_{\boldsymbol{y}}$ is the noise level and $\boldsymbol{X}_0$ and $\boldsymbol{W}$ are independent. The probability of observing $\boldsymbol{Y} = \boldsymbol{y}$ given $\boldsymbol{X}_0 = \boldsymbol{x}_0$ and a degradation operator $\mathcal{A}$ is defined as

$$g_0(\boldsymbol{y}|\boldsymbol{x}_0, \mathcal{A}) := \mathrm{N}\big(y; \mathcal{A}(\boldsymbol{x}_0), \sigma_{\boldsymbol{y}}^2\boldsymbol{I}_{d'}\big). \tag{5}$$

In the Bayesian paradigm, the reconstruction of $\boldsymbol{X}_0$ from an observation $\boldsymbol{Y} = \boldsymbol{y}$ and the known degradation operator $\mathcal{A}$ amounts to sampling from the posterior distribution associated with (4):

$$\pi_0(\boldsymbol{x}_0|\boldsymbol{y}, \mathcal{A}) \propto g_0(\boldsymbol{y}|\boldsymbol{x}_0, \mathcal{A})\, p_0(\boldsymbol{x}_0). \tag{6}$$

**Diffusion posterior sampling.** The core idea of zero-shot diffusion posterior sampling algorithms (Chung et al., 2025; Daras et al., 2024; Janati et al., 2025b) is to approximate the dynamics of a Denoising Diffusion Model (DDM) targeting the posterior distribution (6) by suitably adapting, at inference time, a DDM trained on the prior $p_0$. A key property of these methods is that they require no additional training on data generated from the observation model (4), beyond the training (3) of the DDM model targeting the prior $p_0$. Constructing a DDM for the posterior distribution $\pi_0(\cdot|\boldsymbol{y}, \mathcal{A})$ for a given known $\boldsymbol{y}$ and $\mathcal{A}$ amounts to approximately sampling along the distributional path $(\pi_t(\cdot|\boldsymbol{y}, \mathcal{A}))_{t=0}^T$, defined analogously to the unconditional probability path $(p_t)_{t=0}^T$:

$$\pi_t(\boldsymbol{x}_t|\boldsymbol{y}, \mathcal{A}) := \int q_{t|0}(\boldsymbol{x}_t|\boldsymbol{x}_0)\pi_0(\boldsymbol{x}_0|\boldsymbol{y}, \mathcal{A})\,\mathrm{d}\boldsymbol{x}_0, \tag{7}$$

This is achieved by having a parametric approximation of the posterior denoiser $\mathbb{E}[\boldsymbol{X}_0 \mid \boldsymbol{X}_t = \boldsymbol{x}_t, \boldsymbol{Y} = \boldsymbol{y}]$, where $\boldsymbol{X}_0 \sim p_0$, $Y \sim g_0(\boldsymbol{y}|\boldsymbol{X}_0, \mathcal{A})$, and $\boldsymbol{X}_t \sim q_{t|0}(\cdot|\boldsymbol{X}_0)$. It satisfies the identity (Daras et al., 2024, Eq. 2.15 and 2.17)

$$\mathbb{E}[\boldsymbol{X}_0 \mid \boldsymbol{X}_t = \boldsymbol{x}_t, \boldsymbol{Y} = \boldsymbol{y}] = \mathbb{E}[\boldsymbol{X}_0 \mid \boldsymbol{X}_t = \boldsymbol{x}_t] + \frac{\sigma_t^2}{\alpha_t}\nabla_{\boldsymbol{x}_t}\log\int g_0(\boldsymbol{y}|\boldsymbol{x}_0, \mathcal{A})\, p_{0|t}(\boldsymbol{x}_0|\boldsymbol{x}_t)\,\mathrm{d}\boldsymbol{x}_0, \tag{8}$$

Hence, assuming that we already have pre-trained a neural network $D_{0|t}^\theta(\boldsymbol{x}_t)$ approximating the unconditional denoiser $\mathbb{E}[\boldsymbol{X}_0 \mid \boldsymbol{X}_t = \boldsymbol{x}_t]$, estimating the posterior denoiser $\mathbb{E}[\boldsymbol{X}_0 \mid \boldsymbol{X}_t = \boldsymbol{x}_t, \boldsymbol{Y} = \boldsymbol{y}]$ reduces to approximating the intractable log-gradient on the right-hand side of (8). A widely used approximation (Chung et al., 2023; Ho et al., 2022) replaces $p_{0|t}(\cdot|\boldsymbol{x}_t)$ in the integral by a Dirac mass centered at the prior denoiser $D_{0|t}^\theta(\boldsymbol{x}_t)$, leading to

$$\nabla_{\boldsymbol{x}_t}\log\int g_0(\boldsymbol{y}|\boldsymbol{x}_0, \mathcal{A})\, p_{0|t}(\boldsymbol{x}_0|\boldsymbol{x}_t)\,\mathrm{d}\boldsymbol{x}_0 \approx \nabla_{\boldsymbol{x}_t}\log g_0(\boldsymbol{y}|D_{0|t}^\theta(\boldsymbol{x}_t), \mathcal{A}).$$

**Mixture approximation for guidance.** Among the zero-shot approximations of the right-hand side of (8), the variational posterior sampling MGDM introduced by Janati et al. (2025a) achieves competitive performance. The key idea is to approximate the marginal distribution $\pi_t(\cdot|\boldsymbol{y}, \mathcal{A})$ in (7) by a mixture $\hat{\pi}_t(\cdot|\boldsymbol{y}, \mathcal{A})$ of midpoint-based approximations $\{\hat{\pi}_t^s(\cdot|\boldsymbol{y}, \mathcal{A})\}_{s=1}^{t-1}$, in the spirit of Moufad et al. (2025): $\pi_t(\boldsymbol{x}_t|\boldsymbol{y}, \mathcal{A}) \approx \hat{\pi}_t(\boldsymbol{x}_t|\boldsymbol{y}, \mathcal{A}) := \sum_{s=1}^{t-1} \omega_t^s \hat{\pi}_t^s(\boldsymbol{x}_t|\boldsymbol{y}, \mathcal{A})$, where $(\omega_t^s)_{s=1}^{t-1}$ are non-negative weights summing to 1, and

$$\hat{\pi}_t^s(\boldsymbol{x}_t|\boldsymbol{y}, \mathcal{A}) \propto \left( \int g_0(\boldsymbol{y}|D_{0|s}^\theta(\boldsymbol{x}_s), \mathcal{A}) \, p_{s|t}(\boldsymbol{x}_s|\boldsymbol{x}_t) \, \mathrm{d}\boldsymbol{x}_s \right) p_t(\boldsymbol{x}_t), \tag{9}$$

is an approximation of $\pi_t(\boldsymbol{x}_t|\boldsymbol{y}, \mathcal{A})$ based on a midpoint timestep $s < t$. In Janati et al. (2025a), approximate sampling from the target posterior distribution $\pi_0(\cdot|\boldsymbol{y}, \mathcal{A})$ is performed sequentially through the intermediate distributions $\hat{\pi}_T(\cdot|\boldsymbol{y}, \mathcal{A}), \hat{\pi}_{T-1}(\cdot|\boldsymbol{y}, \mathcal{A}), \ldots, \hat{\pi}_0(\cdot|\boldsymbol{y}, \mathcal{A})$. At each step $t$, sampling from the mixture $\hat{\pi}_t(\cdot|\boldsymbol{y}, \mathcal{A})$ proceeds by first drawing a midpoint index $s < t$ according to the categorical distribution with weights $(\omega_t^s)_{s=1}^{t-1}$, and then sampling from the corresponding component $\hat{\pi}_t^s(\cdot|\boldsymbol{y}, \mathcal{A})$ in (9) using a Gibbs sampler with the following data augmentation scheme:

$$\bar{\pi}_{0,s,t}(\boldsymbol{x}_0, \boldsymbol{x}_s, \boldsymbol{x}_t|\boldsymbol{y}, \mathcal{A}) \propto p_{0|s}(\boldsymbol{x}_0|\boldsymbol{x}_s) \, g_0(\boldsymbol{y}|D_{0|s}^\theta(\boldsymbol{x}_s), \mathcal{A}) \, p_{s|t}(\boldsymbol{x}_s|\boldsymbol{x}_t) \, p_t(\boldsymbol{x}_t). \tag{10}$$

By construction, the $\boldsymbol{x}_t$-marginal of this joint distribution is exactly $\hat{\pi}_t^s(\cdot|\boldsymbol{y}, \mathcal{A})$. The Gibbs sampler (Gelfand & Smith, 1990) repeats for $R$ steps the following updates; given $(\boldsymbol{X}_0^r, \boldsymbol{X}_s^r, \boldsymbol{X}_t^r)$,

1. Sample $\boldsymbol{X}_s^{r+1}$ from

$$\bar{\pi}_{s|0,t}(\boldsymbol{x}_s|\boldsymbol{X}_0^r, \boldsymbol{X}_t^r, \boldsymbol{y}, \mathcal{A}) \propto g_0(\boldsymbol{y}|D_{0|s}^\theta(\boldsymbol{x}_s), \mathcal{A}) \, q_{s|0,t}(\boldsymbol{x}_s|\boldsymbol{X}_0^r, \boldsymbol{X}_t^r). \tag{11}$$

   In practice, this step is implemented using a Gaussian variational distribution, as we describe below. This approximation is precisely what places the method of Janati et al. (2025a) within the class of variational posterior sampling approaches.

2. Sample $\boldsymbol{X}_0^{r+1}$ from $p_{0|s}(\cdot|\boldsymbol{X}_s^{r+1})$ using the approximate reverse process of the pre-trained DDM that targets the prior $p_0$, based on the kernels (2).

3. Sample $\boldsymbol{X}_t^{r+1}$ from $q_{t|s}(\cdot|\boldsymbol{X}_s^{r+1})$, which is a forward Gaussian noising process (1) in DDMs.

**Variational approximation.** Given the context variable $\boldsymbol{c} := (\boldsymbol{x}_0, \boldsymbol{x}_t, s, t, \boldsymbol{y}, \mathcal{A})$, the variational approximation consists in minimizing

$$\min_{\boldsymbol{\mu}, \boldsymbol{\rho}} \mathcal{L}(\boldsymbol{\mu}, \boldsymbol{\rho}; \boldsymbol{c}) := D_{\mathrm{KL}}\left( \mathrm{N}(\cdot; \boldsymbol{\mu}, \mathrm{diag}(\boldsymbol{\rho})) \,\|\, \bar{\pi}_{s|0,t}(\cdot|\boldsymbol{x}_0, \boldsymbol{x}_t, \boldsymbol{y}, \mathcal{A}) \right), \tag{12}$$

where $D_{\mathrm{KL}}$ denotes the Kullback–Leibler divergence, $\bar{\pi}_{s|0,t}(\boldsymbol{x}_s|\boldsymbol{x}_0, \boldsymbol{x}_t, \boldsymbol{y}, \mathcal{A})$ is defined in (11), and $\mathrm{diag}(\boldsymbol{\rho})$ is the diagonal matrix with entries given by the vector $\boldsymbol{\rho}$. This KL divergence is in practice estimated using the reparameterization trick (Kingma & Welling, 2013) with a single Monte Carlo sample. Achieving a sufficiently small value of this objective is critical for obtaining high-quality reconstructions in variational diffusion posterior sampling algorithms (Janati et al., 2025a). However, in practice this minimization is performed at inference time using first-order optimization, which typically requires multiple gradient updates and costly backpropagations through the pre-trained unconditional denoiser $D_{0|t}^\theta$. This high computational cost motivates the amortized optimization strategy introduced in the next section.

## 3 METHOD

We propose an *amortized optimization* approach (Amos, 2023) that transfers the burden of minimizing the variational objective (12) from inference to a dedicated upstream training phase.

Existing amortized variational inference methods with diffusion priors (Feng & Bouman, 2024; Feng et al., 2023; Lee et al., 2024) approximate the posterior $\pi_0(\cdot|\boldsymbol{y}, \mathcal{A})$ by directly learning an implicit distribution to perform one-step inference with a single neural network forward pass. Such an inference does not leverage test-time likelihood guidance like in zero-shot posterior sampling methods, so we hypothesize that these methods are tied to the degradations seen during training and fails under shifted operators, and rely on large quantities of paired clean-and-degraded image data to achieve a good reconstruction quality. In contrast, our method focuses on the amortization of

*inner optimization problems* that arise in variational diffusion posterior sampling algorithms such as MGDM (Section 2). This allows us to have the flexiblity to integrate test-time likelihood guidance on top of amortization, yielding both zero-shot robustness and a significant reduction in inference time. To our knowledge, no prior amortized method offers this combination.

**Inference model for amortized optimization.** Our main contribution is to propose a *fully amortized inference model*, defined as $c \mapsto \lambda_\varphi(c) := (\mu_\varphi(c), \rho_\varphi(c))$, with parameters $\varphi$, to predict a good initialization to address the variational approximation problem (12), given a new context $c$. The model is implemented using a single forward pass through a neural network and trained by minimizing the objective-based loss (Amos, 2023):

$$\min_\varphi \mathbb{E}_c \left[ \mathcal{L}(\mu_\varphi(c), \rho_\varphi(c); c) \right], \tag{13}$$

where $\mathcal{L}$ is the variational objective from (12). The training loss (13) is minimized by stochastic gradient descent. The context $c := (x_0, x_t, s, t, y, \mathcal{A})$ in the expectation of (13) is a random variable drawn from a distribution designed to match those encountered at inference, *i.e.*, during posterior sampling with MGDM. We find the following sampling procedure for $c$ to be effective in practice:

$$(x, y, \mathcal{A}) \sim p_{\text{data}}(x) \, g_0(y|x, \mathcal{A}) \, p_{\text{op}}(\mathcal{A}),$$

$$t \sim \text{Unif}(\mathcal{T}), \quad s \mid t \sim \text{Categorical}\left( (\omega_t^{s'})_{s'=1}^{t-1} \right),$$

$$x_t \mid x \sim q_{t|0}(\cdot|x), \qquad x_0 := D_{0|t}^\theta(x_t).$$

where $p_{\text{data}}(x)$ denotes the distribution of clean images, $p_{\text{op}}(\mathcal{A})$ denotes the distribution of degradation operators modeled during training, and $\mathcal{T} \subseteq \{1, \ldots, T\}$ denotes the subset of timesteps on which the inference model is applied during test-time inference.

**Design of the inference model.** The inference model aims to predict a near minimizer of problem (12), so it is natural to start from the initialization used in zero-shot MGDM (Janati et al., 2025a), where the optimization is initialized with the statistics of the Gaussian bridge transition $q_{s|0,t}(x_s|x_0, x_t) \propto q_{s|0}(x_s|x_0) q_{t|s}(x_t|x_s)$ and then refined by gradient descent. To mimic this procedure in a single forward pass, we parameterize the network so that it learns the *residual* between these prior statistics and the posterior parameters that approximately minimize (12).

Concerning the architecture of this neural network, note that minimizing the variational objective (12) amounts to approximately sampling from the conditional law $\bar{\pi}_{s|0,t}(x_s|x_0, x_t, y, \mathcal{A}) \propto g_0(y|D_{0|s}^\theta(x_s), \mathcal{A}) \, q_{s|0,t}(x_s|x_0, x_t)$ defined in (11), given a context $c = (x_0, x_t, s, t, y, \mathcal{A})$. In other words, the task is to *denoise* from $x_t$ down to an intermediate state $x_s$ with $s < t$, while conditioning on $x_0$ and, crucially, on the observation $y$ and the operator $\mathcal{A}$. This conditioning naturally suggests drawing inspiration from the conditional denoisers used in supervised diffusion models (Saharia et al., 2022a;b), which are expressly designed to handle external inputs such as $y$ and $\mathcal{A}$. Therefore, the neural-network architecture of our inference model can be derived from the backbone of any existing conditional denoisers, with the following minor modifications: (i) concatenate $x_0$, $x_t$ along the channel dimension; (ii) early fusion of the embedding of the timesteps $t$ and $s$; (iii) double the number of output channels so the network jointly predicts variational parameters that approximately minimize (12). These minor changes are compatible with all existing standard conditional denoiser architecture, e.g., UNet (Ho et al., 2020), DiT (Peebles & Xie, 2023), HDiT (Crowson et al., 2024). To efficiently account for multiple degradations during training, we recommend adopting the InvFussion (Elata et al., 2025) architecture, which allows the inference model to handle more than a single degradation operator. Further details are deferred to Appendix B.

**Amortized variational diffusion posterior sampling.** At inference, we propose to use the trained fully-amortized inference model to warm-start the optimization of (12) in MGDM: given a context $c := (x_0, x_t, s, t, y, \mathcal{A})$, it predicts $(\mu_\varphi(c), \rho_\varphi(c))$ in one forward pass to initialize a first-order optimizer for a few gradient steps. This contrasts with zero-shot MGDM (Janati et al., 2025a) that initializes the optimizer from the mean and covariance of the *unconditional* prior bridge transition $q_{s|0,t}(x_s|x_0, x_t) \propto q_{s|0}(x_s|x_0) q_{t|s}(x_t|x_s)$, which completely ignores the conditioning on the observation $y$ and the degradation operator $\mathcal{A}$. We hypothesize that a well-trained inference model provides an initialization tailored to the inverse problem at hand (accounting for $y$ and $\mathcal{A}$) that can

reduce the gradient steps required to minimize the variational objective (12) without compromising the final reconstruction quality in MGDM. Experiments (Section 4) show that such a warm-start-plus-fine-tuning strategy is already competitive, leaving as future work the exploration of more sophisticated amortization methods such as semi-amortized methods (Amos, 2023).

**Related methods.** Two recent works (Elata et al., 2025; Mbakam et al., 2025) share with us the idea of training a component that can be reused at inference to remain flexible to different likelihoods (equivalently, different degradation operators), but they differ in how the likelihood is ultimately exploited: they predict a likelihood-conditioned denoised sample *directly through a network* that takes the likelihood parameters as input. Indeed, Elata et al. (2025) adapt deep unrolling (Gregor & LeCun, 2010) by training a conditional diffusion model with an operator-aware block that applies the degradation operator $\mathcal{A}$ and its pseudo-inverse directly to the activations, enabling the network to predict a likelihood-conditioned denoised sample. Similarly, Mbakam et al. (2025) unfold the LATINO Langevin sampler (Spagnoletti et al., 2025)—a posterior sampler where the likelihood enters each Langevin update through proximal operators— into a finite sequence of learnable steps and train it with a consistency trajectory loss so that the network likewise predicts a likelihood-conditioned denoised sample. Because these methods rely on the network to internalize the likelihood correction, they require large paired datasets and may generalize poorly to degradation operators not covered during training. In contrast, our approach keeps the likelihood-guided optimization explicit: the inference model provides an *approximate* initialization, and the final correction is obtained by directly minimizing the variational objective (12) with the true test-time likelihood. This unique combination of network initialization and explicit likelihood-guidance enables both fast inference and stronger robustness to unseen operators, as demonstrated in the next section. We defer the extended related works to Appendix A.

## 4 EXPERIMENTS

We evaluate the proposed method that we call *amortized MGDM*. The following set of experiments aims to support our three main claims: (i) amortization in MGDM accelerates posterior sampling compared to zero-shot MGDM; (ii) amortized posterior sampling outperforms supervised diffusion when both models are trained in data-poor regimes; (iii) amortized posterior sampling is more robust to OOD degradation operators than the supervised baseline.

**Experimental setting.** Our protocol follows Elata et al. (2025). The unconditional denoiser $D_{0|t}^\theta$ is an HDiT (Crowson et al., 2024) denoiser. In this section, we focus on the FFHQ dataset (Karras et al., 2019). The study on the ImageNet (Deng et al., 2009) dataset is deferred to Appendix E. We also focus here on an image resolution of $64 \times 64$, matching the setting studied in Elata et al. (2025). This choice enables strictly controlled comparisons across methods. Experiments at the larger resolution of $256 \times 256$, which further validate our conclusions, are presented in Appendix D. Methods are evaluated on diverse image restoration tasks: super-resolution, inpainting, and motion deblurring. The noise level in (4) is fixed at[1] $\sigma_{\boldsymbol{y}} = 0.01$. The evaluation is performed on 300 test images, unseen during training or hyperparameter tuning. The reconstruction quality is measured with per-image metrics comparing the similarity between the reconstructed image and a reference one, such as LPIPS (Zhang et al., 2018), SSIM (Wang et al., 2004) and PSNR. We report the average of these metrics with their standard deviations over all test images. To measure the photorealism of the reconstructed images, we also report the CMMD[2] (Jayasumana et al., 2024). Zero-shot MGDM (Janati et al., 2025a) currently sets the state of the art among zero-shot diffusion posterior samplers (Chung et al., 2023; Mardani et al., 2024; Song et al., 2023; Wang et al., 2023; Wu et al., 2024; Zhang et al., 2025; Zhu et al., 2023). We therefore include it as our zero-shot baseline.[3] Palette (Saharia et al., 2022a) and InvFussion (Elata et al., 2025) serve as our supervised diffusion baselines, both relying on a conditional denoiser architecture based on HDiT. Our inference model also adopts an HDiT-based architecture to ensure an apples-to-apples comparison. See Appendix C for details.

---

[1] The chosen value of noise level is slightly lower than the usual $\sigma_{\boldsymbol{y}} = 0.05$ considered in other works.

[2] With only 300 images, we found FID (Heusel et al., 2017) to be unstable and poorly aligned with perceptual quality. We therefore rely on CMMD, which is more sample-efficient and better suited to our experiments.

[3] For completeness, Appendix D.3 compares MGDM with DPS (Chung et al., 2023), analyzing the trade-off between reconstruction quality and inference speed. The results show that MGDM achieves a more favorable balance, motivating our choice to use it as the zero-shot baseline to accelerate via amortization.

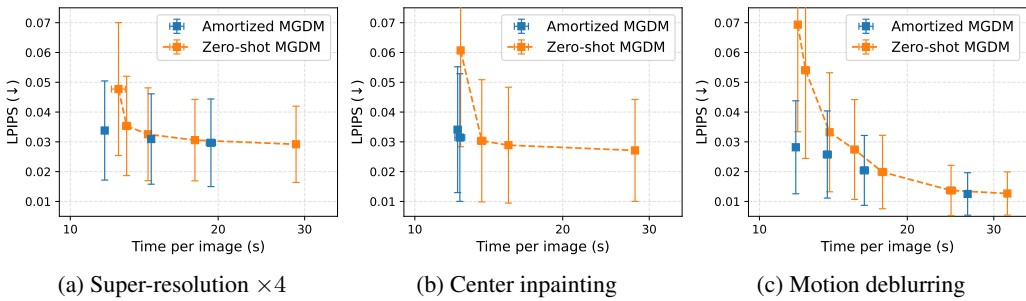

(a) Super-resolution ×4  (b) Center inpainting  (c) Motion deblurring

Figure 1: Trade-off between reconstruction quality (LPIPS) and inference time of amortized MGDM vs. zero-shot MGDM (Janati et al., 2025a). Configurations of zero-shot MGDM are Pareto-optimal.

Table 1: Speedup of amortized MGDM vs. zero-shot MGDM (for a given target LPIPS).

| Metric | Super-resolution ×4 | Inpainting | Motion debluring |
|---|---|---|---|
| Target LPIPS | 0.031 | 0.031 | 0.028 |
| Time (s) (zero-shot→amortized) | $17.9 \to 14.6 \ (\times\mathbf{1.27})$ | $13.7 \to 12.3 \ (\times\mathbf{1.11})$ | $15.6 \to 11.9 \ (\times\mathbf{1.31})$ |

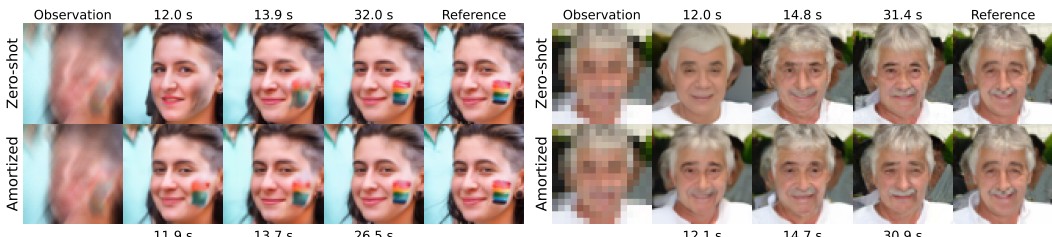

Figure 2: Reconstruction examples of amortized MGDM vs. zero-shot MGDM, on motion deblurring (left) and super-resolution ×4 (right). We report the sampling time.

## 4.1 ACCELERATION OVER ZERO-SHOT POSTERIOR SAMPLING

We demonstrate that amortized MGDM achieves a better trade-off between reconstruction quality and inference speed compared to zero-shot MGDM. In particular, amortization enables faster inference without loss of reconstruction quality.

The inference model is trained following (13) on only 1 % of the pre-training data (600 images), which proves sufficient to achieve competitive performance. At inference, hyperparameter[4] sweeps are used to characterize the trade-off between reconstruction quality (in LPIPS) and inference time, for both amortized and zero-shot MGDM. In this study, the same degradation operator is used for both training and inference across the three inverse problems: super-resolution, inpainting with a fixed mask, and motion deblurring with a fixed kernel. See Appendix C.1 for details on the protocol.

**Results.** Figure 1 and Table 1 quantify the speed advantage at fixed quality (measured in LPIPS). Amortization yields an inference-time speedup ranging from ×1.11 (inpainting) to ×1.31 (super-resolution), while preserving perceptual quality.[5] Appendix D.4 shows that the computational cost of training the inference model is *modest*, requiring only a few hours of training on a single NVIDIA V100 GPU, meaning that the *overall* computational cost (training + inference) of amortized MGDM becomes cheaper once more than a few thousands of images are reconstructed at inference. Figure 2 highlights the quality advantage under an identical inference-time budget: in the low inference-time budget, amortized MGDM produces sharper and more detailed reconstructions. Overall, these results confirm that amortization provides a better trade-off between reconstruction quality and inference speed. Our claim is further supported by the study at the $256 \times 256$ resolution in Appendix D.2, together with the extension to natural images (ImageNet) instead of face images in Appendix E.

---

[4]One important hyperparameter is the number of gradient steps to minimize the variational approximation problem (12). The study of its impact in the quality-speed trade-off is deferred to Appendix D.5.

[5]The equivalence in quality is statistically validated through a paired one-sided t-test on per-image LPIPS scores (see Appendix C.1 for details).

Table 2: Amortized MGDM vs. supervised diffusion Palette (Saharia et al., 2022a) trained with only 1% of the pre-training data. Evaluation on super-resolution ×4 (SR ×4), inpainting (masking a square of size $32 \times 32$ at the center), motion deblurring (fixed kernel), on 300 images of FFHQ64.

| Task | Method | LPIPS ↓ | SSIM ↑ | PSNR ↑ | CMMD ↓ |
|---|---|---|---|---|---|
| SR ×4 | Palette (Saharia et al., 2022a) (1%) | $0.061 \pm 0.035$ | $0.645 \pm 0.060$ | $19.8 \pm 1.6$ | 0.408 |
| | **Amortized MGDM** (1%) | $\mathbf{0.029 \pm 0.016}$ | $\mathbf{0.852 \pm 0.047}$ | $\mathbf{25.4 \pm 2.4}$ | **0.028** |
| Inpainting | Palette (Saharia et al., 2022a) (1%) | $0.040 \pm 0.025$ | $0.904 \pm 0.035$ | $25.9 \pm 3.5$ | 0.126 |
| | **Amortized MGDM** (1%) | $\mathbf{0.025 \pm 0.016}$ | $\mathbf{0.928 \pm 0.025}$ | $\mathbf{27.7 \pm 2.7}$ | **0.015** |
| Motion deblurring | Palette (Saharia et al., 2022a) (1%) | $0.141 \pm 0.054$ | $0.535 \pm 0.083$ | $17.4 \pm 1.7$ | 0.304 |
| | **Amortized MGDM** (1%) | $\mathbf{0.013 \pm 0.007}$ | $\mathbf{0.941 \pm 0.019}$ | $\mathbf{30.5 \pm 2.0}$ | **0.028** |

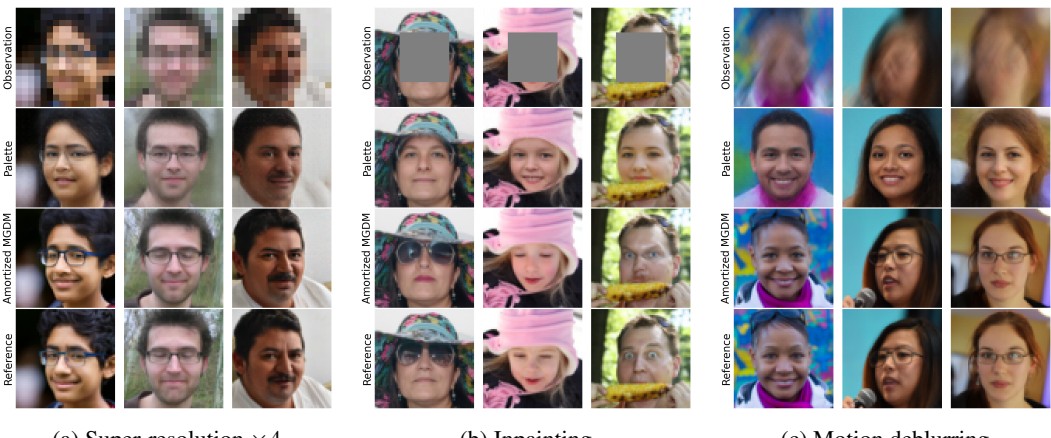

(a) Super-resolution ×4        (b) Inpainting        (c) Motion deblurring

Figure 3: Reconstruction samples comparing amortized MGDM vs. supervised diffusion Palette (Saharia et al., 2022a), when models are trained in a data-scarce regime (1% of data).

## 4.2 OUTPERFORMING SUPERVISED DIFFUSION IN DATA-SCARCE REGIME

Having demonstrated the advantages of the method over zero-shot MGDM, we next compare our method to the supervised diffusion baseline, since amortization requires an additional training phase with paired clean-and-degraded images. While supervised approaches are known to perform well with abundant training data, their behavior in severely data-limited settings remains largely unexplored. To the best of our knowledge, this is the first systematic evaluation of supervised diffusion under such frugal regimes, revealing its vulnerability when training data are scarce.

We again consider here the specific setting where we use the same degradation operator both for inference and training. The amortized inference model in amortized MGDM is trained following (13) using only 1% of the FFHQ dataset (600 images). For comparison, we fine-tune the supervised diffusion baseline Palette[6] (Saharia et al., 2022a) on the same 1% subset. Details of the protocol are provided in Appendix C.2.

**Results.** Table 2 shows that amortized MGDM consistently outperforms the fine-tuned supervised baseline across all metrics and tasks when both models are trained on only 1% of the data. Qualitative examples in Figure 3 further highlight that amortized MGDM produces sharper, more photorealistic reconstructions, whereas the supervised baseline exhibits noticeable blurring. As detailed in Appendix D.6, the performance of the supervised baseline depends on the amount of available training data—its reconstruction quality drops when moving from 10% to 1% of the dataset—while amortized MGDM remains stable. Taken together, these results demonstrate the superior robustness and data efficiency of amortized MGDM over supervised diffusion approaches. Experiments at the higher $256 \times 256$ resolution (Appendix D.6) confirm the same trend.

---

[6]InvFussion (Elata et al., 2025) is omitted, as it targets varying operators while they are fixed in our setting.

Table 3: Amortized MGDM vs. supervised diffusion InvFussion (Elata et al., 2025) on inpainting with out-of-distribution masks (missing pixels). Evaluation on 300 images of FFHQ64. Models were trained on in-distribution masks (single missing rectangle). The best metric is in bold, the second best metric is underlined.

|  | LPIPS ↓ | SSIM ↑ | PSNR ↑ | CMMD ↓ |
|---|---|---|---|---|
| In-distribution masks (missing rectangle) | | | | |
| InvFussion (Elata et al., 2025) | $0.015 \pm 0.027$ | $0.947 \pm 0.075$ | $33.2 \pm 8.1$ | 0.010 |
| Zero-shot MGDM (Janati et al., 2025a) | $\mathbf{0.014 \pm 0.024}$ | $\mathbf{0.952 \pm 0.066}$ | $33.3 \pm 7.1$ | 0.011 |
| **Amortized MGDM** | $\mathbf{0.014 \pm 0.025}$ | $0.949 \pm 0.076$ | $\mathbf{33.5 \pm 7.8}$ | **0.008** |
| Out-of-distribution masks (missing pixels) | | | | |
| InvFussion (Elata et al., 2025) | $0.403 \pm 0.118$ | $0.427 \pm 0.077$ | $16.8 \pm 1.4$ | 1.029 |
| Zero-shot MGDM (Janati et al., 2025a) | $0.040 \pm 0.022$ | $\mathbf{0.846 \pm 0.051}$ | $\mathbf{23.6 \pm 2.5}$ | 0.077 |
| **Amortized MGDM** | $\mathbf{0.039 \pm 0.021}$ | $0.840 \pm 0.053$ | $23.5 \pm 2.6$ | **0.066** |

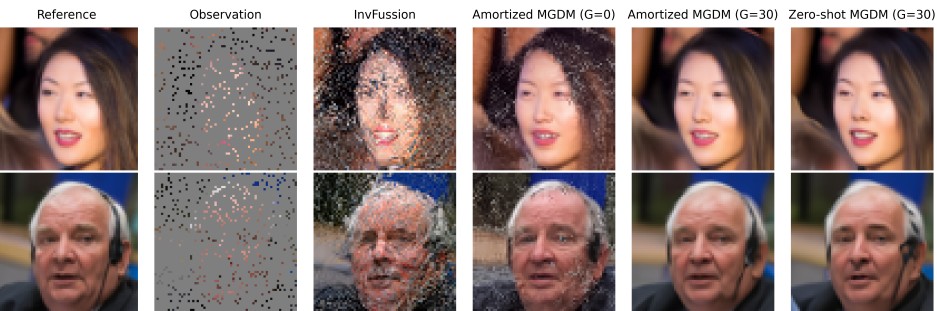

Figure 4: Reconstruction samples for inpainting with OOD masks (missing pixels). Models were trained on in-distribution masks (missing rectangle). $G$: gradient steps (cf. Appendix C.3).

### 4.3 ROBUSTNESS TO OUT-OF-DISTRIBUTION OPERATORS

Our method relies on explicit likelihood guidance, as in posterior sampling, which we show to provide an advantage in robustness to OOD operators compared to InvFussion (Elata et al., 2025)—a supervised diffusion baseline that embeds the degradation operator into the conditional denoiser. While this design is effective when degradations match those seen during training, we hypothesize that it struggles to generalize when the inference-time operator deviates from this distribution. In contrast, amortized MGDM enforces data consistency with the actual operator at inference time, thereby avoiding this generalization bottleneck. It is important to emphasize that our goal is *not* to show that amortized MGDM outperforms zero-shot MGDM under OOD degradations, which would be unrealistic without overlap between the in-distribution and OOD degradation families.

To test our hypothesis, we reproduced the supervised baseline InvFussion (Elata et al., 2025) following its exact setup for image inpainting, considering no more a fixed mask, but different families of masks during training and inference. During training, the learned conditional denoiser only observes masks that remove a single randomly placed rectangle (in-domain degradations). The inference model of amortized MGDM is trained on the same family of in-distribution masks. At evaluation, we introduce a different family of masks in which individual pixels are randomly removed according to an i.i.d. Bernoulli sampling with parameter $p \in [0.9, 0.92]$, creating OOD degradations. Details are deferred to Appendix C.3.

**Results.** Table 3 shows that the supervised baseline InvFussion (Elata et al., 2025) performs well on in-domain masks but collapses under OOD masks, producing strong artifacts visible in Figure 4, despite explicitly encoding the degradation operator. This illustrates its lack of robustness when the inference-time degradation departs from what was modeled during training.

In contrast, amortized MGDM achieves strong reconstruction quality on in-domain masks, and—importantly—amortization does not degrade OOD performance relative to the zero-shot base-

line. As expected, the warm start provided by an inference model trained only on in-domain operators is weaker when applied to OOD degradations: with no additional gradient steps ($G = 0$), amortized MGDM may fail and produce artifacts, although these remain less severe than those observed with InvFussion (cf. Figure 4). However, once we introduce gradient steps to minimize the variational objective (12) using the actual OOD operator, the method reliably recovers the same performance as zero-shot MGDM. This verification is both necessary and nontrivial. In principle, warm-starting variational inference from an in-domain model could bias optimization on OOD degradations and yield worse results than zero-shot MGDM. Our experiments show that this does not happen.

Together, these findings highlight the central role of test-time likelihood guidance in achieving robustness to OOD degradations. They also illustrate the limitations of supervised baselines, which fail when the degradation operator is unseen during training. An additional experiment in Appendix D.7 involving a complete operator shift—training on motion deblurring and testing on super-resolution—at the higher $256 \times 256$ resolution further reinforces this conclusion.

## 5 CONCLUSION

We presented *amortized variational diffusion posterior sampling*, a diffusion framework that bridges supervised diffusion and zero-shot posterior sampling by pairing an upstream amortized training phase with a test-time posterior-sampling procedure that explicitly incorporates likelihood guidance. This design achieves faster inference than pure zero-shot posterior sampling while remaining robust in data-scarce training settings and adaptable to OOD degradation operators.

A current limitation is that reconstruction quality on in-distribution operators does not surpass a well-trained supervised baseline when large paired datasets are available. However, amortized MGDM proves more effective in low-data regimes (Section 4.2), and, unlike the supervised baseline, leverages test-time likelihood guidance to maintain robustness to OOD degradations (Section 4.3).

Future work could explore how to better leverage large-scale paired datasets during the upstream training phase to further close the gap with supervised diffusion on in-domain operators, while maintaining the strong OOD robustness demonstrated in this work. Another promising direction is to investigate refined amortization schemes—such as semi-amortization in the spirit of Marino et al. (2018)—to further reduce inference cost or improve reconstruction quality beyond full amortization in variational diffusion posterior sampling.

## REPRODUCIBILITY STATEMENT

For the sake of reproducible research, we open-source our code upon acceptance. Important details to help reproducibility of the experimental results are gathered in Appendix C.

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

## A    EXTENDED RELATED WORK

Our diffusion posterior sampling approach uses amortized optimization within variational diffusion posterior sampling algorithms (Janati et al., 2025a; Moufad et al., 2025). This is in contrast to methods that perform variational inference directly with a diffusion prior, as described below. Since our method requires training data for amortization, we compare it to supervised diffusion baselines and thus provide a comparison with diffusion posterior sampling algorithms, which to our knowledge has been largely under-researched in previous work.

**Amortized optimization.**    Amortized optimization replaces repeated solutions of an optimization problem $\min_{\boldsymbol{\lambda}} \mathcal{L}(\boldsymbol{\lambda}; \boldsymbol{c})$, parameterized by a context variable $\boldsymbol{c}$, by a learned mapping $\boldsymbol{c} \mapsto \boldsymbol{\lambda}_{\varphi}(\boldsymbol{c})$, implemented by a neural network with parameters $\varphi$ (Amos, 2023). The central idea is to exploit statistical regularities across contexts observed during training so that the network directly predicts $\boldsymbol{\lambda}$ and provides an effective approximate minimizer of $\mathcal{L}(\cdot; \boldsymbol{c})$ without performing iterative optimization from scratch. A canonical example of this paradigm is the amortized variational inference (Kingma & Welling, 2013).

**Variational inference with diffusion prior.**    Previous work (Feng & Bouman, 2024; Feng et al., 2023; Lee et al., 2024; Mammadov et al., 2024) has investigated the acceleration of posterior sampling for inverse problems with diffusion priors using amortized variational inference. These methods train offline to learn a mapping from the observation to an implicit distribution that approximates the posterior, implemented either with normalizing flows (Feng & Bouman, 2024; Feng et al., 2023; Mammadov et al., 2024) or with neural networks that push forward a Gaussian distribution (Lee et al., 2024).

While these approaches enable fast posterior sampling, bypassing the iterative backward diffusion of zero-shot or supervised conditional diffusion methods, their practical effectiveness remains uncertain. They require paired, clean-and-degraded training data and have not been systematically compared with supervised diffusion baselines (Elata et al., 2025; Saharia et al., 2022a;b).

In contrast, our method uses amortized optimization to accelerate the sampling of specific backward transition kernels in variational diffusion posterior sampling algorithms (Janati et al., 2025a; Moufad et al., 2025). In addition to reducing sampling costs, our approach retains the flexibility to incorporate the likelihood at inference time which proves to be particularly beneficial when training data is limited, or when facing OOD degradation operators.

**Diffusion posterior sampling vs. supervised diffusion.**    Supervised diffusion methods (Elata et al., 2025; Saharia et al., 2022a;b) use an end-to-end trained conditional denoiser and directly simulate the backward diffusion process using the learned conditional denoiser during inference. These approaches generally outperform zero-shot diffusion posterior sampling (Chung et al., 2023; Janati et al., 2025a) when the degradation operator is explicitly modeled during training and sufficient training data is available, as shown in Elata et al. (2025). There are however a few instances of zero-shot methods outperforming supervised methods; in Wu et al. (2024), the zero-shot method FRAMEDIFF is able to outperform RFDIFFUSION (Watson et al., 2023) on some motif-scaffolding benchmarks, and in Moufad et al. (2025), the MGPS algorithm is able to outperform a supervised diffusion baseline on the task of electrocardiogram lead inpainting.

We hypothesize that the effectiveness of supervised methods diminishes in data-scarce regimes. To our knowledge, our work is the first systematic investigation of this scenario. Our results highlight the limitations of supervised diffusion in such situations and emphasize the advantages of diffusion posterior sampling methods that incorporate the degradation operator at inference time. This feature provides flexibility at test time, a key feature of the proposed method.

## B    ADDITIONAL DETAILS ON THE METHOD (SECTION 3)

We recall that the context variable $\boldsymbol{c} := (\boldsymbol{x}_0, \boldsymbol{x}_t, s, t, \boldsymbol{y}, \mathcal{A})$ in (12) is a tuple containing images $\boldsymbol{x}_0$ and $\boldsymbol{x}_t$ at time 0 and $t$, the two timesteps $t$ and $s$, the observation $\boldsymbol{y}$ and the degradation operator $\mathcal{A}$.

**Parameterization.** We parameterize the inference model using a neural network $\boldsymbol{c} \mapsto \boldsymbol{f}_\varphi(\boldsymbol{c}) := (\boldsymbol{f}_\varphi^{\mathrm{mean}}(\boldsymbol{c}), \boldsymbol{f}_\varphi^{\mathrm{var}}(\boldsymbol{c}))$ that predicts the residual between the mean, covariance statistics $(\boldsymbol{\mu}_{s|0,t}(\boldsymbol{x}_0, \boldsymbol{x}_t), \sigma_{s|0,t}^2 \boldsymbol{I}_d)$ of the prior Gaussian transition $q_{s|0,t}(\boldsymbol{x}_s|\boldsymbol{x}_0, \boldsymbol{x}_t) \propto q_{s|0}(\boldsymbol{x}_s|\boldsymbol{x}_0)q_{t|s}(\boldsymbol{x}_t|\boldsymbol{x}_s)$, and the target posterior statistics $(\boldsymbol{\mu}_\varphi(\boldsymbol{c}), \boldsymbol{\rho}_\varphi(\boldsymbol{c}))$ that warm-start the variational approximation problem (12):

$$\boldsymbol{\mu}_\varphi(\boldsymbol{c}) := \boldsymbol{\mu}_{s|0,t}(\boldsymbol{x}_0, \boldsymbol{x}_t) + \boldsymbol{f}_\varphi^{\mathrm{mean}}(\boldsymbol{c}),$$

$$\boldsymbol{\rho}_\varphi(\boldsymbol{c}) := \sigma_{s|0,t}^2 \mathbf{1}_d + \boldsymbol{f}_\varphi^{\mathrm{var}}(\boldsymbol{c}),$$

where $\mathbf{1}_d$ denotes the vector full of ones,

$$\boldsymbol{\mu}_{s|0,t}(\boldsymbol{x}_0, \boldsymbol{x}_t) := \gamma_{t|s}\alpha_{s|0}\boldsymbol{x}_0 + (1 - \gamma_{t|s})\alpha_{t|s}^{-1}\boldsymbol{x}_t \qquad (14)$$

with $\gamma_{t|s} := \sigma_{t|s}^2/\sigma_{t|0}^2$, and

$$\sigma_{s|0,t}^2 := \sigma_{t|s}^2\sigma_{s|0}^2/\sigma_{t|0}^2, \qquad (15)$$

because by definition, $q_{s|0,t}(\boldsymbol{x}_s|\boldsymbol{x}_0, \boldsymbol{x}_t) \propto q_{s|0}(\boldsymbol{x}_s|\boldsymbol{x}_0)q_{t|s}(\boldsymbol{x}_t|\boldsymbol{x}_s)$, and $q_{s|0}(\boldsymbol{x}_s|\boldsymbol{x}_0) := \mathrm{N}\left(\boldsymbol{x}_s; \alpha_{s|0}\boldsymbol{x}_0, \sigma_{s|0}^2\boldsymbol{I}_d\right)$ and $q_{t|s}(\boldsymbol{x}_t|\boldsymbol{x}_s) := \mathrm{N}\left(\boldsymbol{x}_t; \alpha_{t|s}\boldsymbol{x}_s, \sigma_{t|s}^2\boldsymbol{I}_d\right)$, by definition (1).

The choice of this parameterization is motivated by the fact the zero-shot MGDM (Janati et al., 2025a) proposes to initialize the variational approximation problem (12) at inference time using the prior statistics $\boldsymbol{\mu}_{s|0,t}(\boldsymbol{x}_0, \boldsymbol{x}_t)$ and $\sigma_{s|0,t}^2$, which are updated using gradient descent updates rules.

**Architecture.** The architecture of the neural network model $\boldsymbol{c} \mapsto \boldsymbol{f}_\varphi(\boldsymbol{c}) := (\boldsymbol{f}_\varphi^{\mathrm{mean}}(\boldsymbol{c}), \boldsymbol{f}_\varphi^{\mathrm{var}}(\boldsymbol{c}))$ is inspired by the standard design of conditional denoisers used in supervised diffusion. Such denoisers take as inputs the noised image $\boldsymbol{x}_t$, the timestep $t$, the observation $\boldsymbol{y}$ and the degradation operator $\mathcal{A}$, and output a prediction of the denoised image (Elata et al., 2025; Saharia et al., 2022a;b). In our setting, the neural network model involved in the parameterization of the inference model requires two additional inputs: the midpoint timestep $s < t$ and the condition on the reconstructed image $\boldsymbol{x}_0$ at timestep 0. Its outputs are the residual $\boldsymbol{f}_\varphi^{\mathrm{mean}}(\boldsymbol{c})$ of the approximate Gaussian distribution and the residual diagonal entries $\boldsymbol{f}_\varphi^{\mathrm{var}}(\boldsymbol{c})$ of its covariance, both having the same dimensionality as an image. These minor changes are compatible with all existing standard conditional denoiser architecture, e.g., UNet, DiT, HDiT, etc.

There are many possible modifications of the standard conditional denoiser to accommodate these changes in inputs and outputs. We found the following design to be effective in practice: (i) concatenate $\boldsymbol{x}_0$ and $\boldsymbol{x}_t$ channel-wise, thereby doubling the input channels; (ii) embed the midpoint timestep $s$ analogously to the embedding of $t$, but independently (e.g., using a separate linear layer), and add both timestep embeddings at the beginning of the architecture (early fusion); (iii) double the number of output channels to jointly predict the Gaussian mean and diagonal covariance entries.

Following the design of the conditional denoisers proposed in Elata et al. (2025), the backbone of the neural network is an InvFussion conditional denoiser, which is an HDiT (Crowson et al., 2024) involving a so-called feature degradation layer that incorporates the degradation operator into the architecture, so that the architecture can adapt to different degradation operators during training and inference. In the experiments where a same degradation operator is fixed during training and inference, we simplify the architecture by removing these feature degradation layers.

**Training algorithm and design of the context distribution.** The training algorithm of the inference model is summarized in Algorithm 1. We now comment the choice of the context distribution involved in the training objective function (13) for amortization. The sampling of $(\boldsymbol{x}, \boldsymbol{y}, \mathcal{A})$ creates a tuple of a clear image, its degraded observation along with the corresponding degradation operator, following the model (4). The sampling of the timesteps $(t, s)$ with $s < t$ aims to match those encountered during posterior sampling with MGDM, cf. Section 2. Finally, the choice of the distribution for $(\boldsymbol{x}_t, \boldsymbol{x}_0)$ aims to capture the fact that, during posterior sampling with MGDM, $\boldsymbol{x}_t$ is a noised sample, and $\boldsymbol{x}_0$ is a denoised version of $\boldsymbol{x}_t$: for this reason, we choose to set $\boldsymbol{x}_0 := D_{0|t}^\theta(\boldsymbol{x}_t)$ rather $\boldsymbol{x}_0 := \boldsymbol{x}$, where $\boldsymbol{x}$ is sampled directly from $p_{\mathrm{data}}$, the distribution of clean images.

**Inference algorithm.** Algorithm 2 summarizes how the trained inference model is used during our method called amortized MGDM. The three conditional distributions involved in the Gibbs sampling of Section 2, as described below (10), are implemented in Algorithm 2 as follows:

---

**Algorithm 1** Training fully-amortized inference model

---

**Require:** Dataset of clean images $\mathcal{D}$, distribution of operators $p_{\text{op}}$, subset $\mathcal{T} \subseteq \{1, \ldots, T\}$ of timesteps, learning rate $\alpha > 0$
**Ensure:** Trained inference model $\boldsymbol{\lambda}_\varphi$
1: $\varphi \leftarrow$ initialization from scratch
2: **for** each iteration **do**
3:     Sample clean images $\boldsymbol{x} \sim \mathcal{D}$
4:     Sample degradation operator $\mathcal{A} \sim p_{\text{op}}$
5:     Sample observation $\boldsymbol{y} \sim g_0(\boldsymbol{y}|\boldsymbol{x}, \mathcal{A})$
6:     Sample timestep $t \sim \text{Unif}(\mathcal{T})$
7:     Sample midpoint timestep $s \sim \text{Categorical}\left((\omega_t^{s'})_{s'=1}^{t-1}\right)$,
8:     Sample $\boldsymbol{x}_t \sim q_{t|0}(\cdot|\boldsymbol{x})$                    {Forward noising}
9:     $\boldsymbol{x}_0 \leftarrow D_{0|t}^\theta(\boldsymbol{x}_t)$                    {Pre-trained unconditional denoiser}
10:    $\boldsymbol{c} \leftarrow (\boldsymbol{x}_0, \boldsymbol{x}_t, s, t, \boldsymbol{y}, \mathcal{A})$
11:    Compute gradient $g \leftarrow \nabla_\varphi \mathcal{L}(\boldsymbol{\mu}_\varphi(\boldsymbol{c}), \boldsymbol{\rho}_\varphi(\boldsymbol{c}); \boldsymbol{c})$
12:    Update parameters $\varphi \leftarrow \varphi - \alpha g$
13: **end for**
14: **return** Inference model $\boldsymbol{\lambda}_\varphi := (\boldsymbol{\mu}_\varphi(\boldsymbol{c}), \boldsymbol{\rho}_\varphi(\boldsymbol{c}))$

---

- lines 7–16 perform the sampling of $\boldsymbol{x}_s$ conditioned on $\boldsymbol{x}_0$ and $\boldsymbol{x}_t$ via a Gaussian variational approximation, using a certain number of gradient steps;

- line 17 handles the sampling of $\boldsymbol{x}_0$ given $\boldsymbol{x}_s$, e.g., through a DDIM sampler (Song et al., 2021a) with $M$ steps following the pre-trained unconditional denoiser $D_{0|t}^\theta$;

- line 18 addresses the sampling of $\boldsymbol{x}_t$ conditioned on $\boldsymbol{x}_0$, using a Gaussian forward noising process.

## C   EXPERIMENTAL PROTOCOL FOR SECTION 4

This section regroups all the important details to reproduce the results in Section 4.

**Inference model training.**   The inference model used in amortized MGDM is trained by minimizing the population loss (13) with $1\%$ of the FFHQ dataset (600 images), using AdamW (Loshchilov & Hutter, 2019) with a learning rate of $10^{-4}$, a weight decay of $10^{-5}$, a batch size of 32, during 200, 300 or 500 epochs on a single V100. All these training hyperparameters were tuned on a validation loss. The subset of timesteps $\mathcal{T} \subseteq \{1, \ldots, T\}$ that is chosen in practice is of the form $\mathcal{T} := \{1, \ldots, \lceil(1 - r_{\text{switch}})T\rceil\}$, for a certain parameter $r_{\text{switch}} \in [0, 1]$ that we tune.

**Hyperparameters of MGDM.**   We describe the hyperparameters of Algorithm 2 that we consider in our experiments. In our experiments with MGDM, we set the weights $(\omega_t^s)_{s=1}^{t-1}$ of the approximation mixture $\pi_t(\boldsymbol{x}_t|\boldsymbol{y}, \mathcal{A}) \approx \hat{\pi}_t(\boldsymbol{x}_t|\boldsymbol{y}, \mathcal{A}) := \sum_{s=1}^{t-1} \omega_t^s \hat{\pi}_t^s(\boldsymbol{x}_t|\boldsymbol{y}, \mathcal{A})$ (see Section 2) to $\omega_t^s = 0$ for $s < t - 1$ and $\omega_t^{t-1} = 1$, as this configuration proved most effective in practice. We always choose a number of $K = 100$ timesteps when running MGDM, and the number of Gibbs repetition is $R = 1$. The timesteps $(t_k)_{k=0}^K$ is a uniform grid between $t_0 = 0$ and $t_K = T$. The choice of the subset of timesteps $\mathcal{T} \subseteq \{1, \ldots, T\}$ for which we apply the inference model is the same as the one considered during training: $\mathcal{T} := \{1, \ldots, \lceil(1 - r_{\text{switch}})T\rceil\}$, for a certain parameter $r_{\text{switch}} \in [0, 1]$ that we tune. The number of gradient steps $(G_k)_{k=1}^K$ are chosen as follow:

$$G_k := \begin{cases} G_{\text{start}} & \text{if } t_k \notin \mathcal{T}, \\ G_{\text{end}} & \text{otherwise,} \end{cases}$$

where $G_{\text{start}}, G_{\text{end}}$ are hyperparameters to tune. The number of denoising steps $M$ is tuned as well. We use the Adam optimizer (Kingma & Ba, 2015) without weight decay to minimize the variational objective function (12), with a learning rate $\eta$ to tune.

---

**Algorithm 2** Amortized variational diffusion posterior sampling

---

**Require:** Likelihood $g_0(\boldsymbol{y}|\cdot, \mathcal{A})$ defined from (5) with observation $\boldsymbol{y}$ and operator $\mathcal{A}$, unconditional denoiser $(\boldsymbol{x}, t) \mapsto D_{0|t}^\theta(\boldsymbol{x})$ pre-trained via (3), inference model $\boldsymbol{c} \mapsto \boldsymbol{\lambda}_\varphi(\boldsymbol{c})$ trained via (13), number of timesteps K, timesteps $(t_k)_{k=0}^K$ with $t_K = T$ and $t_0 = 0$, Gibbs repetition $R$, number of denoising steps $M$, learning rates $\eta$, gradient steps $(G_k)_{k=1}^K$, probabilities $(\omega_k^\ell)_{k \in \{2, \ldots, K-1\}, \ell \in \{1, \ldots, t_{k-1}\}}$, subset of timesteps $\mathcal{T} \subseteq \{1, \ldots, T\}$ for which we apply the inference model

**Ensure:** Approximate sample of the posterior distribution $\pi_0(\cdot|\boldsymbol{y}, \mathcal{A})$ defined in (6)
1: $\boldsymbol{x}_t \sim \mathcal{N}(0, \boldsymbol{I}_d)$
2: $\boldsymbol{x}_0 \leftarrow D_{0|T}^\theta(\boldsymbol{x}_t)$
3: **for** $k = K - 1 \rightarrow 2$ **do**
4:    $s \sim \text{Categorical}((\omega_k^\ell)_{\ell=1, \ldots, t_{k-1}})$
5:    $\boldsymbol{x}_t \sim q_{t_k|0, t_{k+1}}(\cdot|\boldsymbol{x}_0, \boldsymbol{x}_t)$
6:    **for** $r = 1 \rightarrow R$ **do**
7:      $\boldsymbol{c} \leftarrow (\boldsymbol{x}_0, \boldsymbol{x}_t, s, t_k, \boldsymbol{y}, \mathcal{A})$
8:      **if** $t_k \in \mathcal{T}$ **then**
9:        $(\boldsymbol{\mu}, \boldsymbol{\rho}) \leftarrow \boldsymbol{\lambda}_\varphi(\boldsymbol{c}) := (\boldsymbol{\mu}_\varphi(\boldsymbol{c}), \boldsymbol{\rho}_\varphi(\boldsymbol{c}))$              {Warm-start with inference model}
10:      **else**
11:        $(\boldsymbol{\mu}, \boldsymbol{\rho}) \leftarrow (\boldsymbol{\mu}_{s|0, t_k}(\boldsymbol{x}_0, \boldsymbol{x}_t), \sigma_{s|0, t_k}^2 \mathbf{1}_d)$, cf. (14) and (15)
12:      **end if**
13:      **for** $g = 1 \rightarrow G_k$ **do**
14:        $(\boldsymbol{\mu}, \boldsymbol{\rho}) \leftarrow (\boldsymbol{\mu}, \boldsymbol{\rho}) - \eta \nabla_{(\boldsymbol{\mu}, \boldsymbol{\rho})} \mathcal{L}(\boldsymbol{\mu}, \boldsymbol{\rho}; \boldsymbol{c})$                   {Likelihood-guidance}
15:      **end for**
16:      $\boldsymbol{x}_s \sim \mathcal{N}(\boldsymbol{\mu}, \text{diag}(\boldsymbol{\rho}))$                          {Sampling $\boldsymbol{x}_s$ knowing $\boldsymbol{x}_0$ and $\boldsymbol{x}_t$}
17:      $\boldsymbol{x}_0 \leftarrow \text{DDIM}(\boldsymbol{x}_s, s, M)$                              {Sampling $\boldsymbol{x}_0$ knowing $\boldsymbol{x}_s$}
18:      $\boldsymbol{x}_t \sim q_{t_k|s}(\cdot|\boldsymbol{x}_s)$                               {Sampling $\boldsymbol{x}_t$ knowing $\boldsymbol{x}_s$}
19:    **end for**
20: **end for**
21: **return** $\boldsymbol{x}_0$

---

**Time measurements.** All runtimes are measured on a single NVIDIA V100 GPU with batch size 1. For each image, we time the full sampling loop of the chosen method, using Python's high-resolution wall clock (`time.perf_counter`). Timing starts immediately before the sampling call and stops right after it returns; thus, preprocessing (degradation setup, noise generation), metric computation, and file I/O are excluded.

**Image reconstruction tasks.** We describe the degradation operators corresponding to each of the three considered reconstruction tasks: super-resolution, inpainting and motion deblurring. We rely on the implementations of degradation operators provided in the codebase of Elata et al. (2025); Janati et al. (2025a).

- **Super-resolution:** The degradation operator is a uniform downsampling operation, by averaging over non-overlapping patches of size $r \times r$. We set $r = 4$, corresponding to the task of $\times 4$ super-resolution.

- **Inpainting:** The degradation operator is a masking operator, where only pixels corresponding to unmasked indices are retained, and the inverse problem is to reconstruct the full image. We consider the following masks. Center inpainting (for Section 4.1 and Section 4.2): we remove a central square region covering half of the image ($I/2 \times I/2$ pixels for an image of size $I \times I$). Missing rectangle (for Section 4.3): we choose to mask a rectangle of random shape and random position, as implemented by Elata et al. (2025). Missing pixels (for Section 4.3): we choose to mask some randomly selected pixels, following the implementation of MissingPatch in Elata et al. (2025).

- **Motion deblurring:** The degradation operator is a convolution with a fixed motion blurring kernel. We use a kernel of size $21 \times 21$ with intensity 0.9, following the implementation of MotionBlur in Janati et al. (2025a).

**Training, validation test dataset.** The image indices of the FFHQ dataset considered for training the unconditional denoiser, the conditional denoisers and the inference model and those ranging from 0 to 59999 included. When training on a subset of $1\%$ of the images, we consider the indices ranging from 0 to 599. The image indices for our validation set are those ranging from 61000 to 61031 included. The image indices for our test set (final evaluation of all methods) are those ranging from 69000 to 69299.

**Normalization of pixel values.** To give a meaning to the relative noise level $\sigma_{\boldsymbol{y}}$ in (4), we assume that the pixel values are normalized between $-1$ and $1$.

### C.1 EXPERIMENTS ON THE INFERENCE TIME ACCELERATION WITH AMORTIZATION (SECTION 4.1)

For the zero-shot MGDM baseline, we perform an extensive hyperparameter search on 32 validation images, evaluating more than 140 combinations to determine the Pareto front of reconstruction quality (measured with LPIPS, a perceptual similarity metric well aligned with human visual judgments) versus inference time. This Pareto front represents the set of configurations that yield the best trade-off between reconstruction fidelity and inference speed for zero-shot MGDM. Using the same hyperparameter grid, we tune amortized MGDM to identify configurations surpassing this Pareto front.

**Hyperparameter grid.** To study the trade-off between reconstruction quality and inference speed of MGDM in both zero-shot and amortized settings, we explore a broad grid of more than 140 tuples of hyperparameters. We follow the naming conventions introduced in Algorithm 2. The considered grid is:

- $G_{start} \in \{1, 3, 10\}$
- $G_{end} \in \{0, 1, 3, 10\}$
- Learning rate $\eta \in \{0.01, 0.03\}$
- Number of denoising steps $M \in \{1, 5\}$
- Switch ratio $r_{\text{switch}} \in \{70\%, 80\%, 90\%\}$

**Pareto front construction.** We trace the zero-shot Pareto front by sweeping the hyperparameters presented previously on a 32-image validation set, thereby characterizing the trade-off between reconstruction quality and inference time. We then take the best zero-shot configurations (highest quality at a given speed) and evaluate it on 300 images from the test set.

**Statistical test.** We compare amortized MGDM to the zero-shot baseline using a paired non-inferiority $t$-test on per-image LPIPS scores (denoted by $\text{LPIPS}_{\text{ZS}}$ for the zero-shot baseline, and $\text{LPIPS}_{\text{AM}}$ for amortized MGDM, lower is better). For each test image $i$, we form the paired difference

$$\delta_i := \text{LPIPS}_{\text{ZS},i} - \text{LPIPS}_{\text{AM},i},$$

so that larger $\delta_i$ favors amortized MGDM.

With non-inferiority margin $\Delta > 0$, the hypotheses are

$$H_0 : \ \delta \leq -\Delta \qquad \text{(amortized MGDM worse than zero-shot MGDM by at least } \Delta\text{)},$$
$$H_1 : \ \delta > -\Delta \qquad \text{(amortized is not inferior to zero-shot MGDM)}.$$

Let $\bar{\delta}$ and $\bar{s}$ be the empirical mean and standard deviation of the samples $\{\delta_i\}_{i=1}^n$. The statistical test is

$$T \ = \ \frac{\bar{\delta} + \Delta}{\bar{\delta}/\sqrt{n}},$$

We reject $H_0$ at level $\alpha$ if

$$T > t_{1-\alpha,\, n-1},$$

where $t_{1-\alpha,\,n-1}$ denotes the $(1-\alpha)$-quantile of the Student $t$ distribution with $n-1$ degrees of freedom. Equivalently, we conclude non-inferiority if the one-sided $(1-\alpha)$ lower confidence bound for $\delta$ exceeds $-\Delta$.

We use $\alpha = 0.05$ and $\Delta = 0.003$. Configurations of amortized MGDM that reject $H_0$ are considered to be configurations that does not degrade the reconstruction quality compared to the considered zero-shot MGDM sampling. They are therefore retained for speedup reporting when aiming for a certain target quality, as it is done in Table 1.

### C.2 Experiments where the training dataset is scarce (Section 4.2)

We tune the hyperparameters of amortized MGDM and the the supervised diffusion baseline Palette on a validation set of 32 images from `FFHQ64`, and select the hyperparameters yielding the smallest average LPIPS. The fine-tuning of the conditional denoiser is initialized from the pre-trained unconditional model.

We use the same grid of hyperparameters as in the previous subsection to tune amortized MGDM on the validation set. We reuse the code from Elata et al. (2025) to implement the supervised baseline Palette (Saharia et al., 2022a). The conditional denoiser is initialized from the weights of the unconditional denoiser and fine-tuned with the Adam optimizer for 16384 steps using a batch size of 512. We use a constant learning rate of $5 \times 10^{-5}$ with no weight decay. At inference, we employ the DDIM sampler with 500 steps to generate reconstructions with the supervised baseline Palette.

### C.3 Experiments on the robustness to OOD masks in inpainting (Section 4.3)

**Training and inference of InvFussion.** The training of the supervised baseline InvFussion (Elata et al., 2025) uses the full training set of `FFHQ64` (60000 images), following the exact protocol of Elata et al. (2025). In their protocol, they use an Adam optimizer without weight decay, with a reference learning rate of $5 \times 10^{-5}$ that follows a warm-up-and-decaying schedule. The model is trained on 32768 kimgs with a batch size of 512. At inference, we use the DDIM sampler with 500 steps to generate reconstructions with the supervised diffusion baseline InvFussion (Elata et al., 2025).

**Configuration for amortized MGDM.** The inference model for amortized MGDM is trained on only $10\%$ of `FFHQ64` (6000 images) for 3 epochs, with an AdamW optimizer. We used a constant learning rate of $10^{-4}$, a weight decay of $10^{-5}$, and a batch size of 32. At inference, the hyperparameters of amortized MGDM are the following: we use $K = 100$ diffusion steps, the number of gradient steps are $G_{\text{start}} = 3$ and $G_{\text{end}} = G$ with $G \in \{0, 30\}$ gradient updates applied to the variational approximation during the final 10 steps, *i.e*, for timesteps $t_k$ with $k \in \{1, \ldots, 10\}$. The learning rate $\eta$ for the variational approximation problem is set to 0.01. The number of Gibbs repetition is $R = 1$, and the number of denoising steps is $M = 1$.

## D Additional experiments on `FFHQ256`

This section extends our experimental study from Section 4 to a resolution of $256 \times 256$, demonstrating that our claims continue to hold at larger image resolutions.

### D.1 Experimental protocol

**Dataset, tasks.** We consider the `FFHQ256` dataset. The inverse problem that we consider is motion deblurring, with a kernel size of $21 \times 21$ and an intensity of 0.9, as well as super-resolution $\times 4$ (where the degradation is the average pooling downsampling). The noise level on the observation is $\sigma_{\boldsymbol{y}} = 0.01$. Following the protocol of InvFussion (Elata et al., 2025), we train from scratch an HDiT (Crowson et al., 2024), scaled to operate on $256 \times 256$ images, using 256000 gradient steps, a learning rate of $5 \times 10^{-5}$, and a batch size of 256.

**Supervised diffusion baseline.** The supervised baseline considered is Palette (Saharia et al., 2022a). The conditional diffusion models are trained using either $1\%$ or $10\%$ of paired clean–

degraded images from the `FFHQ256` dataset. Their weights are initialized from the corresponding unconditional models, after which they are fine-tuned via Adam (Kingma & Ba, 2015) with a constant learning rate of $10^{-5}$ for up to 64,000 gradient steps. The optimal number of fine-tuning steps is treated as a hyperparameter and selected based on validation performance.

Inference with the supervised diffusion baseline uses DDIM (Song et al., 2021a). For each number of sampling steps $T \in \{100, 200, 500, 1000\}$, we tune $\eta \in \{0, 0.25, 0.5, 0.75, 1\}$ on a validation set of 32 images and keep the value giving the best performance. This yields, for each $T$, a configuration optimized for its inference-time budget.

**Training the inference model.**    The inference model for amortized MGDM is trained from scratch using either $1\%$ or $10\%$ of paired clean–degraded images from the `FFHQ256` dataset, to minimize the population loss (13). The optimizer is AdamW (Loshchilov & Hutter, 2019), with a constant learning rate fixed at $10^{-4}$, a weight decay of $10^{-5}$ and a batch size of 16. Training on $1\%$ of the dataset (600 images) is run for 300 epochs, while training on $10\%$ of the dataset (6000 images) is run for 50 epochs.

### D.2    ACCELERATION OF AMORTIZED MGDM VS. ZERO-SHOT MGDM

We extend the experiments from Section 4.1 to motion deblurring and super-resolution on `FFHQ256`, where we compare the trade-off between reconstruction quality and inference speed of amortized MGDM vs. the zero-shot MGDM baseline.

**Protocol.**    To properly study the trade-offs between speed and reconstruction quality, we evaluate both methods on the same hyperparameter grid as in Appendix C.1, using the validation set (32 images). We first identify the Pareto-optimal hyperparameters for zero-shot MGDM, where reconstruction quality is measured by the average LPIPS. For each target quality level on this Pareto front, we then determine the hyperparameters for which amortized MGDM provides a speedup without degrading reconstruction quality. Specifically, we consider that amortized MGDM does not degrade quality with respect to a given zero-shot configuration if it rejects the one-sided statistical test $H_0$ introduced in Appendix C.1. All selected hyperparameter settings (for both amortized and zero-shot MGDM) are subsequently evaluated on the test set (300 images).

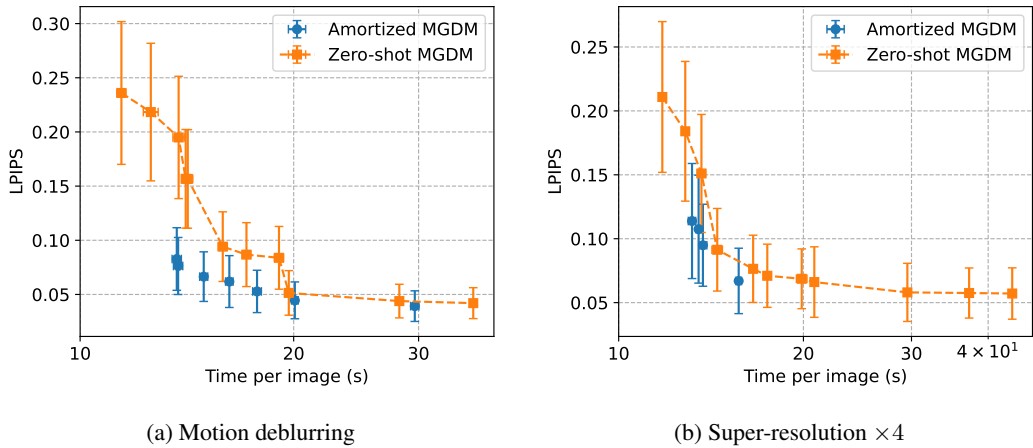

(a) Motion deblurring

(b) Super-resolution $\times 4$

Figure 5: Trade-off between reconstruction quality (LPIPS) and inference time of amortized MGDM vs. zero-shot MGDM, in `FFHQ256`. Configurations of zero-shot MGDM are Pareto-optimal.

**Results.**    Figure 5 confirms that the amortization with the inference model yields an acceleration of inference for amortized MGDM compared to the zero-shot MGDM baseline, without degrading the reconstruction quality. This means that the empirical observations in resolution $64 \times 64$ in Section 4.1 still hold in the larger resolution $256 \times 256$. For instance:

- In motion deblurring, for a target LPIPS of $0.044$, amortized MGDM has a speedup of $\times 1.4$ compared to zero-shot MGDM ($20.5$ vs. $28.7$ seconds per image).

- In super-resolution $\times 4$, for a target LPIPS of $0.066$, amortized MGDM has a speedup of $\times 1.32$ compared to zero-shot MGDM ($15.7$ vs. $20.8$ seconds per image).

## D.3 COMPARISON WITH OTHER ZERO-SHOT POSTERIOR SAMPLING METHODS

We put into perspective the trade-off between reconstruction quality and inference time of MGDM studied in Appendix D.2 compared to other existing zero-shot posterior sampling algorithm, such as DPS (Chung et al., 2023).

**Protocol.** The zero-shot DPS baseline is tuned on the validation set (32 images). We vary the number of sampling steps ($T \in 10, 30, 100, 300, 1000$) and the step sizes ($\zeta \in 0.1, 0.3, 1, 3$), and retain for each $T$ the best-performing configuration. The final evaluation is conducted on the test set (300 images). This procedure yields the reconstruction–time trade-off for DPS, which we compare against the zero-shot MGDM trade-off studied in Appendix D.2.

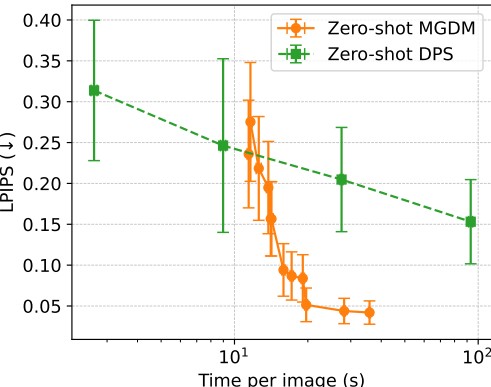

Figure 6: Trade-off between reconstruction quality (LPIPS) and inference time of zero-shot MGDM (Janati et al., 2025a) vs. DPS (Chung et al., 2023), for motion deblurring in `FFHQ256`. Configurations of MGDM and DPS are Pareto-optimal.

**Results.** Figure 6 shows that zero-shot MGDM consistently achieves a better trade-off between reconstruction quality and inference time than DPS. Across the full range of tested compute budgets, MGDM obtains lower LPIPS at equal inference time. For example, for inference times below 30 seconds, DPS reaches an average LPIPS of $0.205$, whereas zero-shot MGDM attains a substantially lower LPIPS of $0.044$. Even when allowing longer runtimes, DPS only improves to an average LPIPS of $0.153$ at 98 seconds of inference.

These results are consistent with prior observations that MGDM provides higher reconstruction fidelity than other zero-shot sampling methods (Janati et al., 2025a).

## D.4 COMPUTATIONAL COST TO TRAIN THE INFERENCE MODEL IN AMORTIZED MGDM

We show that the computational cost required to train the inference model in amortized MGDM is relatively modest compared to the cost of fine-tuning the supervised baseline. In practice, this training overhead is quickly compensated by the substantial speedup achieved during inference relative to the zero-shot MGDM baseline.

**Protocol.** We compare three settings—supervised baseline, zero-shot MGDM, and amortized MGDM—under configurations selected to match a target reconstruction quality corresponding to a certain average LPIPS (here we choose $0.066$ for illustration). For the supervised baseline, we select the fastest sampling configuration that meets this target. This corresponds to DDIM sampling

with 100 diffusion timesteps and $\eta = 0$. For zero-shot MGDM and amortized MGDM, we choose inference hyperparameters that achieve an LPIPS lower than the target value. Specifically, we use $G_{\text{end}} = 10$, $G_{\text{start}} = 1$, $M = 1$, a learning rate of 0.03, and $r_{\text{switch}} = 80\%$ for zero-shot MGDM, while amortized MGDM uses the same settings except for $r_{\text{switch}} = 90\%$. Both the inference model and the supervised baseline are trained on $10\%$ of `FFHQ256`.

Table 4: Training time, inference time (average) and LPIPS (average and standard error) for zero-shot MGDM, amortized MGDM and Palette (Saharia et al., 2022a), on motion deblurring in FFHQ256. Metrics at inference are computed over $300$ test images.

| Method | Training time (GPUh) | Inference time / image (s) | LPIPS ↓ |
|---|---|---|---|
| Zero-shot MGDM | 0 | 28.7 | $0.044 \pm 0.018$ |
| Amortized MGDM (10%) | 6.6 | 20.5 | $0.045 \pm 0.017$ |
| Palette (Saharia et al., 2022a) (10%) | 112.5 | 5.4 | $0.066 \pm 0.020$ |

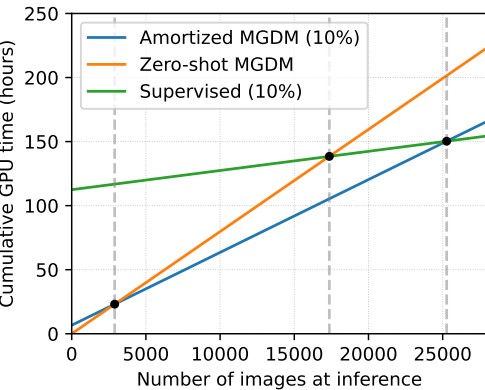

Figure 7: Total computational cost (training cost + inference cost) vs. the number of images to reconstruct at inference, for zero-shot and amortized MGDM, and for the supervised baseline Palette (Saharia et al., 2022a), for motion deblurring in `FFHQ256`.

**Results.** Table 4 shows that training the inference model in amortized MGDM requires only 6.6 GPU-hours on a single GPU, which is substantially lower than the cost of fine-tuning the supervised baseline (14.1 hours on eight V100 GPUs, i.e., 112.45 GPU-hours in total). Figure 7 plots the total computational cost—expressed in GPU-hours—as a function of the number of reconstructed images, according to total_cost = training_cost + $N \times$ inference_cost_per_image, where $N$ denotes the number of inference-time reconstructions. Because the supervised baseline amortizes its substantial training cost over inference, it only becomes cheaper than amortized MGDM after roughly $N = 25{,}260$ images. For number of images $N$ between approximately 2,900 and 25,260 images, amortized MGDM is strictly more computationally efficient when taking into account to total cost (training + inference), compared to the zero-shot or the supervised baselines.

These results clarify the end-to-end efficiency trade-off: a small amount of amortization training accelerates MGDM inference time with a speedup of $\times 1.4$, while requiring over an order of magnitude less total GPU time than supervised diffusion.

### D.5 IMPACT OF THE NUMBER OF GRADIENT STEPS IN AMORTIZED AND ZERO-SHOT MGDM

We analyze how the number of inference gradient steps, denoted $G_{\text{end}}$, affects both reconstruction quality and inference speed for amortized and zero-shot MGDM.

**Protocol.** MGDM hyperparameters are fixed as follows: switch ratio $r_{\text{switch}} = 80\%$; the number of gradient steps for timesteps outside $\mathcal{T} := \{1, \ldots, \lceil (1 - r_{\text{switch}})T \rceil \}$ is $G_{\text{start}} = 1$; the learning rate is $\eta = 0.03$; and the number of denoising steps is $M = 1$. We then vary the number of gradient

steps $G_{end} \in \{0, 1, 3, 10\}$ timesteps in $\mathcal{T}$, to study its role in both amortized and zero-shot MGDM. We considered here 32 images from the validation set.

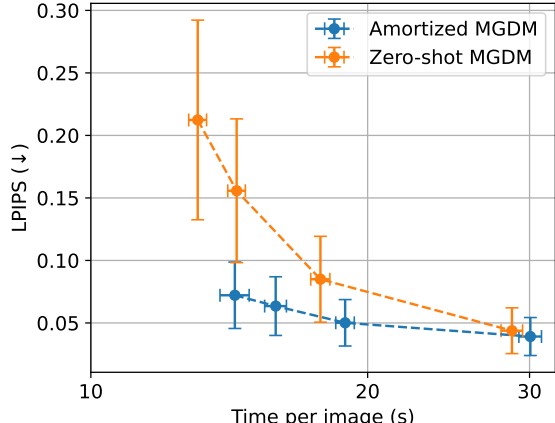

| $G_{end}$ | Zero-shot MGDM | Amortized MGDM |
|---|---|---|
| 0 | $0.212 \pm 0.080$ | $0.072 \pm 0.027$ |
| 1 | $0.156 \pm 0.057$ | $0.063 \pm 0.023$ |
| 3 | $0.085 \pm 0.034$ | $0.050 \pm 0.019$ |
| 10 | $0.044 \pm 0.018$ | $0.039 \pm 0.015$ |

Figure 8: Impact of the number of gradient steps $G_{end}$ on reconstruction performance and inference time, for motion deblurring on `FFHQ256` Left: LPIPS vs. inference time for amortized and zero-shot MGDM. Right: corresponding LPIPS values (mean $\pm$ std) for the different values of number of gradient steps $G_{end}$.

**Results.** As shown in Figure 8, *for the same choice of hyperparameter*, amortized MGDM is slightly slower than zero-shot MGDM due to the additional forward pass through the inference model used to warm-start the variational problem. However, this small overhead is outweighed by a substantial improvement in reconstruction quality, confirmed quantitatively in Figure 8. Overall, amortized MGDM maintains high reconstruction quality even with very few gradient steps, whereas zero-shot MGDM performs poorly in the regime with a low number of gradient steps. This highlights that the learned inference model enables amortized MGDM to operate with significantly fewer gradient steps (hence, a lower inference time) while preserving performance, leading to a better quality-time trade-off.

### D.6   IMPACT OF THE NUMBER OF TRAINING SAMPLES

We study how the number of paired clean–degraded samples affects the performance of both the amortized MGDM inference model and the conditional diffusion model used in the supervised diffusion baseline.

**Protocol.** The hyperparameters of each method (supervised baseline and amortized MGDM) are tuned on a validation set of 32 images, using average LPIPS as the selection metric. For the supervised baseline, we use the hyperparameter grid described in Appendix D.1. For amortized MGDM, we use the grid reported in Appendix C.1, extended with the additional value $G_{end} = 30$. Final performance is measured on 300 test images.

**Results.** As shown in Tables 5 and 6, with only 1% training data, amortized MGDM achieves strictly better reconstruction quality than the supervised diffusion baseline for motion deblurring across all metrics (LPIPS, SSIM, PSNR, CMMD), and yields better LPIPS and CMMD for super-resolution $\times 4$. This result supports our central claim: amortized MGDM can outperform supervised methods in low-data regimes.

When increasing the training data to 10% in motion deblurring, the supervised baseline achieves a lower LPIPS, indicating that its perceptual quality improves with more data. However, amortized MGDM still outperforms it on SSIM, PSNR, and CMMD. This highlights a key difference between the two approaches: (i) the supervised baseline strongly benefits from additional training data and relies on larger datasets to reach high performance; (ii) amortized MGDM remains robust even with limited supervision, making it preferable when annotated data is scarce.

Table 5: Amortized MGDM vs. supervised diffusion baseline Palette (Saharia et al., 2022a), on super-resolution $\times 4$ in `FFHQ256`, trained on $1\%$ and $10\%$ of the data.

| Method | Time (s) ↓ | LPIPS ↓ | SSIM ↑ | PSNR ↑ | CMMD ↓ |
|---|---|---|---|---|---|
| Palette (Saharia et al., 2022a) (1%) | **54.48 ± 0.60** | 0.059 ± 0.020 | **0.867 ± 0.043** | **30.5 ± 2.4** | 0.111 |
| Amortized MGDM (1%) | 62.77 ± 1.08 | **0.054 ± 0.021** | 0.862 ± 0.042 | 30.3 ± 2.4 | **0.042** |

Table 6: Amortized MGDM vs. supervised diffusion baseline Palette (Saharia et al., 2022a), on motion deblurring in `FFHQ256`, trained on $1\%$ and $10\%$ of `FFHQ256`.

| Method | Time (s) ↓ | LPIPS ↓ | SSIM ↑ | PSNR ↑ | CMMD ↓ |
|---|---|---|---|---|---|
| Palette (Saharia et al., 2022a) (1%) | **54.48 ± 0.60** | 0.038 ± 0.014 | 0.901 ± 0.026 | 32.8 ± 2.0 | 0.204 |
| Amortized MGDM (1%) | 85.92 ± 1.76 | **0.030 ± 0.011** | **0.937 ± 0.015** | **35.3 ± 1.8** | **0.077** |
| Palette (Saharia et al., 2022a) (10%) | **54.48 ± 0.60** | **0.023 ± 0.008** | 0.922 ± 0.021 | 34.2 ± 1.9 | 0.109 |
| Amortized MGDM (10%) | 73.63 ± 0.83 | 0.030 ± 0.012 | **0.936 ± 0.016** | **35.2 ± 1.8** | **0.072** |

### D.7 OOD EVALUATION: FROM MOTION DEBLURRING TO SUPER-RESOLUTION

We extend the OOD robustness results from Section 4.3 to a more challenging setting, where the gap between the training and inference operators is significantly larger. Instead of varying mask families within the inpainting task, we now consider entirely different tasks, training on motion deblurring and evaluating at inference time on super-resolution.

**Protocol.** The supervised baseline Palette (Saharia et al., 2022a) is trained on $10\%$ of `FFHQ256`, and evaluated using the setup from Appendix D.6. Amortized MGDM uses an inference model trained on the same data fraction and with identical hyperparameters as in Appendix D.6. Both models are trained on the motion-deblurring operator (kernel size $21 \times 21$, intensity 0.09) and evaluated on the $\times 4$ super-resolution task.

Table 7: OOD generalization from motion deblurring (training operator) to $\times 4$ super-resolution (inference operator). Both models are trained on $10\%$ of `FFHQ256` using only the motion-deblurring operator.

| | LPIPS ↓ | SSIM ↑ | PSNR ↑ | CMMD ↓ |
|---|---|---|---|---|
| **In-distribution operator (motion deblurring)** | | | | |
| Palette (Saharia et al., 2022a) | **0.023 ± 0.008** | 0.922 ± 0.021 | 34.2 ± 1.9 | 0.109 |
| Amortized MGDM | 0.030 ± 0.012 | **0.936 ± 0.016** | **35.2 ± 1.8** | **0.072** |
| **Out-of-distribution operator (super-resolution $\times 4$)** | | | | |
| Palette (Saharia et al., 2022a) | 0.777 ± 0.054 | 0.168 ± 0.065 | 12.78 ± 0.99 | 3.623 |
| Amortized MGDM | **0.087 ± 0.031** | **0.855 ± 0.048** | **29.38 ± 2.72** | **0.151** |

**Results.** As reported in Table 7, the supervised baseline Palette shows poor performance in the OOD setting, with low perceptual and reconstruction metrics. Amortized MGDM, however, achieves noticeably better LPIPS, SSIM, PSNR, and CMMD scores despite the mismatch between the training and inference operators. This indicates that MGDM maintains a meaningful level of generalization to a new inverse problem even when trained on a different task.

## E    ADDITIONAL EXPERIMENTS ON NATURAL IMAGES (IMAGENET)

This section extends our experimental study to natural images from the ImageNet (Deng et al., 2009) dataset, demonstrating that our claim in Section 4.1 on the inference acceleration of amortized MGDM with respect to zero-shot MGDM continue to hold beyond face images.

**Protocol.** We consider the ImageNet dataset at resolution $64 \times 64$, called `IN64` in the following. The inverse problem we study is $4\times$ super-resolution, where the degradation is average-pool downsampling. The observation noise level is $\sigma_y = 0.01$. Following the protocol of InvFussion (Elata et al., 2025), we train an HDiT (Crowson et al., 2024) from scratch, scaled to operate on $64 \times 64$ images, using $256,000$ gradient steps, a learning rate of $5 \times 10^{-5}$, and a batch size of $512$. For validation, we select one image per class from the first 32 ImageNet classes, taking the first image within each class. The test set consists of the last image from each of the first 300 ImageNet classes. We apply classifier-free guidance with a scale of 2 when using the network in both MGDM and AMGDM (Ho & Salimans, 2023).

The inference model for amortized MGDM is trained from scratch using $0.1\%$ of paired clean–degraded images from the `IN64` dataset, in order to minimize the population loss (13). The optimizer is AdamW (Loshchilov & Hutter, 2019), with a constant learning rate set to $5 \times 10^{-5}$ and a batch size of 32. Training on $0.1\%$ of the dataset (1,282 images) is performed for 150 epochs.

Finally, the protocol to study the trade-off between reconstruction quality and inference time for amortized MGDM and zero-shot MGDM follows exactly the one described in Appendix D.2.

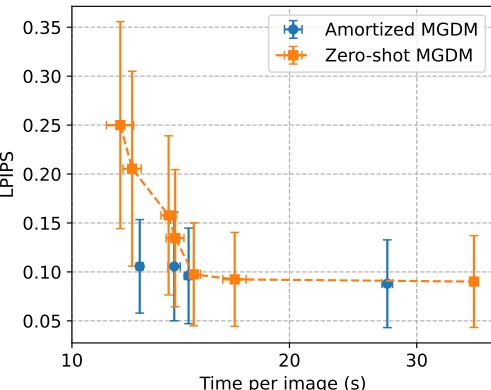

Figure 9: Trade-off between reconstruction quality (LPIPS) and inference time of amortized MGDM vs. zero-shot MGDM, for super-resolution $\times 4$ on `IN64`. Configurations of zero-shot MGDM are Pareto-optimal.

**Results.** Figure 9 confirms that amortizing MGDM with a learned inference model accelerates inference compared with the zero-shot MGDM baseline, while preserving reconstruction quality. For instance, for a target LPIPS of $0.134$, the inference time of amortized MGDM is $12.4$ seconds, yielding a speedup of $\times 1.12$ compared to the $13.9$ seconds for the zero-shot baseline. This demonstrates that the acceleration gains previously observed on face images in FFHQ are not dataset-specific: they extend to the more diverse ImageNet dataset with natural images. This further validates the robustness and scalability of the amortized inference mechanism across datasets and visual domains.

## USE OF LARGE LANGUAGE MODELS

We used large language models solely to help polish the writing, which does not constitute a contribution significant enough to merit authorship credit.

