# OpenReview forum: "The Best of Both Worlds: Amortized Variational Diffusion Posterior Sampling"
_ICLR.cc/2026/Conference — Submitted to ICLR 2026_

### Official Review · Reviewer_WWNq · 2025-10-27

**Soundness:** 2
**Presentation:** 3
**Contribution:** 2
**Rating:** 2
**Confidence:** 4

**Summary:**

This paper approaches the problem of inverse problem inference using diffusion priors for image reconstruction tasks. The authors base their approach on the recent paper: A mixture-based framework for guiding diffusion models, which introduces the MGDM approach. This approach involves optimizing a variational approximation to the posterior of a sample given a higher noise-level sample and a partial observation, as part of a larger Gibbs-sampling approach to sampling at a given time conditioned on a partial observation. The authors propose training an amortization network to initialize this optimization, potentially saving significant computation over the more naive initialization used in the original work. The authors validate their approach with a number of experiments on the FFHQ dataset.

**Strengths:**

**Method**

- The authors provide a clear motivation for the approach, and it seems reasonable that the computational expense of the MGDM approach  would necessitate improvements such as this.
- While the novelty of this approach is somewhat limited, it seems like a reasonable approach that I would expect might have the claimed benefits.
The design of the amortization network appears to be thoughtful and sound

**Writing**
- The paper is clearly written and provides a good introduction to diffusion models for images and Bayesian inverse problems, as well as the quite complex MGDM approach

**Weaknesses:**

**Evaluation setup**

Evaluation is not up to contemporary standards for the following reasons:

- Only a single dataset is considered: FFHQ. FFHQ has significantly less image diversity than natural image datasets such as imagenet, widely used in the literature.
- Only models at a 64x64 resolution are considered, which is well below the 256x256 or higher benchmarks present in other papers.
- The base model used is non-standard, it is not from the original HDiT paper, nor is it a commonly used foundation model such as Guided diffusion, LDMs or stable diffusion.
- Comparison is only against MGDM for running time and Palette for supervised diffusion. A larger set of comparisons would be needed.

All of these issues make it largely impossible to determine the real-world practically of this method and whether the claimed results would scale to larger, more modern diffusion models.

**Performance**

The primary benefit seems to be speeding up inference compared to the Zero-Shot MGDM approach, which could be useful. However, I have significant concerns about the usefulness of this method in general:

- Runtime is only compared to the MGDM and not to other methods. While there is an improvement, >10s still seems like a long time for inference on a 64x64 image and there are no other benchmarks to put this into perspective.
- This method does require significant training for the amortization network, making it only useful over the true zero-shot approach if many similar reconstructions need to be performed.
- The experiments showing OOD performance are unconvincing. While it does appear to outperform InvFussion as predicted, the performance and runtime appears equivalent to the zero-shot MGDM approach, still performing poorly with no gradient updates.

**Novelty**

The novelty is somewhat limited as it does not introduce a fundamentally new approach to inverse problem inference, rather applies amortization to speed up an existing approach.

**Questions:**

My understanding is that even with amortization the approach still requires potentially multiple steps of gradient descent for each of R Gibbs sampling steps within each step of denoising. How does this compare in practice to other methods for zero-shot inference?

---

> ### Author Response · Authors · 2025-11-22
>
> Thank you for your review. We added new experiments in **Appendix D** to address the feedbacks of all reviewers. We now respond to your comments specifically.
>
> ### Regarding your question
>
> > The approach still requires gradient-descent steps within each Gibbs step. How does this compare to other zero-shot methods?
>
> **Appendix D.3** now compares MGDM with DPS [1]. As in the original MGDM paper [2], we use $R = 1$ Gibbs step. On FFHQ-256 motion deblurring, zero-shot MGDM achieves a better quality–time trade-off:
> |Inference Time|Method|LPIPS|
> |-|-|-|
> |30 s|DPS|0.205|
> ||Zero-shot MGDM |**0.044**|
> |98 s|DPS|0.153|
>
> Thus MGDM is not slower than DPS for comparable quality.
>
> ### Regarding the other points you mentioned
>
> 1. > Only 64×64 resolution is considered.
>
> The revision now includes a new set of experiments in **Appendix D** showing that the claims of the paper holds at the larger $256 \times 256$ image resolution, beyond the original $64 \times 64$ setting.
>
> We initially focused on $64 \times 64$ to enable a controlled comparison where all methods (zero-shot MGDM, amortized MGDM, supervised diffusion) share the same training budgets and evaluation protocol.
>
> 2. > The base model is non-standard.
>
> The HDiT model [3] is used in prior inverse problems work (e.g., InvFussion [4]). Our implementation matches the original architecture of HDiT.
>
> The amortization mechanism is architecture-agnostic. Building an inference model from any existing architecture backbone for conditional denoiser only requires: (i) adding output channels to predict mean + variance in the variational approximation; (ii) adding input channels for conditioning on $(x_0, x_t)$; (iii) fusing the timestep embedding of $s$ and $t$.
>
> 3. > Comparison only against MGDM for running time and Palette for supervised diffusion.
>
> We focus on MGDM because it is among the strongest zero-shot methods [2]. **Appendix D.3** now adds a comparison with DPS for completeness.
>
> For supervised methods, we evaluate Palette [7] and InvFussion [4], both competitive baselines. The conditional denoisers are pretrained and then fine-tuned on 1–10% paired data, with hyperparameter tuning to ensure competitiveness (see **Appendix C and D.1**).
>
>
> 4. > Unclear whether the method scales to modern diffusion models.
>
> We agree that results on FFHQ-64 do not guarantee scalability to larger modern diffusion models. However, our experiments are carried out under a rigorous evaluation protocol, e.g., fair comparison with hyperparameter tuning. We additionally extend experiments to $256 \times 256$ (**Appendix D**), providing preliminary evidence of scalability.
>
> 5. > Training amortization is costly and useful only when many reconstructions are needed.
>
> **Appendix D.4** shows that the inference model trains in 6.6 GPU-hours on a single V100, which is not costly compared, e.g., to the cost of training the supervised diffusion baseline. The table below summarizes total computational cost (motion deblurring on FFHQ-256, 10% data):
> |Method|Train time (GPU h)| Inference time / image (s)|LPIPS|
> |-|-|-|-|
> |Zero-shot MGDM|0|28.7|0.044 $\pm$ 0.018|
> |Amortized MGDM (10%)|6.6|20.5|0.045 $\pm$ 0.017|
>
> When taking into account the total computation cost (training + inference), amortized MGDM becomes cheaper for $N \geq 2898$ reconstructions and gives a 1.4× inference speedup.
>
> 6. > OOD experiments are unconvincing.
>
> The goal is not to outperform zero-shot MGDM on OOD degradations—this is unrealistic when the inference model is trained only on in-domain operators. The key question is whether restricting training to in-domain degradations harms OOD performance relative to zero-shot MGDM.
>
> **Section 4.3** shows that it does not: the inference network, despite being trained solely on in-domain operators, still provides a reasonable initialization, where adding subsequent gradient steps recover the same quality as zero-shot MGDM. In contrast, supervised diffusion collapses on OOD operators.
>
> 7. > Limited novelty.
>
> While amortization has been considered before [5,6], our method is novel because it introduces amortization *within each diffusion step*. This design leads to a combination of properties that, to our knowledge, is not offered by prior work: (i) an upstream training phase enabling faster inference compared to zero-shot approaches, and (ii) the ability to incorporate test-time likelihood guidance, which improves robustness in OOD settings.
>
> ### References
>
> [1] Diffusion posterior sampling for general noisy inverse problems, 2023
>
> [2] A Mixture-Based Framework for Guiding Diffusion Models, 2025
>
> [3] Scalable high-resolution pixel-space image synthesis with hourglass diffusion transformers, 2024
>
> [4] InvFussion: Bridging Supervised and Zero-shot Diffusion for Inverse Problems, 2025
>
> [5] Diffusion prior-based amortized variational inference for noisy inverse problems, 2024
>
> [6] Amortized posterior sampling with diffusion prior distillation, 2025.
>
> [7] Palette: Image-to-image diffusion models, 2022

---

### Official Review · Reviewer_WbUn · 2025-10-31

**Soundness:** 2
**Presentation:** 1
**Contribution:** 2
**Rating:** 4
**Confidence:** 5

**Summary:**

The paper tackles an important gap between two existing paradigms for diffusion‐based inverse problems: (i) zero‑shot variational diffusion posterior sampling (VDPS), which is robust but slow; and (ii) supervised diffusion models, which are fast but require large paired datasets and often fail under unseen degradations. The proposed amortized variational diffusion posterior sampling (Amortized MGDM) trains an inference model on a small set of paired data to predict a good initialization for the variational approximation in VDPS. By retaining explicit likelihood guidance during test time, the method aims to combine the adaptability of VDPS with the speed of supervised models. This is a creative application of amortized optimization to diffusion posterior sampling.

**Strengths:**

- Clarity: The paper provides a clear derivation of the approach. It describes how the inference network takes the current noisy sample, the clean reference, the timesteps, the degraded observation and operator as input and outputs residual mean and variance to initialize the variational optimization. The use of residual prediction relative to the unconditional bridge transition and architecture modifications (concatenating x_0 and x_t, summing timestep embeddings, doubling channels) are explained. The method maintains explicit likelihood guidance via subsequent gradient steps, ensuring the final reconstruction remains consistent with the degradation operator.

- Significance: Accelerating variational diffusion posterior sampling while retaining robustness has practical value for inverse problems. Demonstrating that modest training (1 % of data) plus explicit likelihood guidance can outperform fully supervised diffusion models at data-poor scenarios highlights a compelling alternative for certain scenarios.

**Weaknesses:**

- Inaccurate baseline discussion and limited comparison: The paper’s discussion of prior amortized variational inference methods is inaccurate. The authors argue that previous amortized inference approaches with diffusion priors (e.g., [1], [2]) require paired degraded–clean datasets, which is not the case. These methods can learn implicit priors using only degraded data without access to ground-truth clean samples. This misunderstanding should be properly addressed, as it leads to an unfair dismissal of related methods. Moreover, the paper’s experimental comparison is incomplete: it only contrasts Amortized MGDM with zero-shot MGDM, Palette, and InvFussion, omitting several recent works that also aim to accelerate posterior sampling. For instance, [1] adopts amortized variational posterior sampling and can solve inverse problems in a single step, directly targeting the same problem setup. It also demonstrates strong OOD performance. A careful quantitative and conceptual comparison with such approaches is necessary to situate the contribution accurately within the existing literature.

- Limited speed‑up: Although the authors report speed gains of roughly 10–32 % over zero‑shot MGDM, the improvements are modest compared to supervised diffusion models that require only a single network pass. Amortized MGDM still involves training an inference network and then performing gradient steps during inference; for OOD operators the paper notes that around 30 additional steps were necessary to match zero‑shot performance. Thus the method does not eliminate the test‑time optimization burden and may remain slower than purely supervised approaches. Moreover, Table 2 compares reconstruction quality between Amortized MGDM and a supervised baseline but does not report inference time, limiting a fair assessment of the trade‑offs. Without discussing the cost of the total inference time and performance vs. supervised methods, it is difficult to gauge practical advantages.

- Data requirements and generality: The method requires data to train the inference model. While the authors successfully use only 600 images (1 % of pre‑training data) for FFHQ64, it is unclear how much data would be needed for higher‑resolution tasks (e.g., ImageNet) or more complex modalities. The evaluation is restricted to 64×64 images from FFHQ; there are no experiments on higher resolutions, other datasets, or other modalities (e.g., medical imaging), so the generality of the approach remains uncertain.

- Dependence on hyperparameters: The inference network is derived from a specific HDiT denoiser architecture. Adapting the approach to other diffusion backbones like UNet or DiT may require non‑trivial modifications. Performance depends on the number of gradient steps and hyperparameter tuning for both zero‑shot and amortized MGDM. Sensitivity analyses for the warm‑start network capacity, data size, and number of gradient steps are absent, leaving it unclear how robust the method is to these choices.

---
[1] Diffusion Prior-Based Amortized Variational Inference for Noisy Inverse Problems, ECCV 2024 \
[2] AMORTIZED POSTERIOR SAMPLING WITHDIFFUSION PRIOR DISTILLATION, ICLR 2025 workshop

**Questions:**

- Have the authors compared Amortized MGDM against recent amortized inference approaches with diffusion priors? These methods often learn implicit priors and can perform single-step inference (e.g., DAVI). A direct comparison and breakdown of performance versus the inference cost would clarify the methodological and empirical differences between Amortized MGDM and prior amortized diffusion inference models.

- Missing FID evaluation: Have the authors tried to compare the FID performance of Amortized MGDM agains existing baselines?

- Gradient step sensitivity: How does reconstruction quality and inference time vary with the number of gradient steps following the amortized warm-start? Is there a clear trade-off between speed and quality, and is there a principled or data-dependent way to select the optimal number of steps?

- Have you tested Amortized MGDM on higher-resolution datasets such as ImageNet-256 × 256 or other domains (e.g., natural images, medical data)? Demonstrating generalization across resolutions and domains is crucial for establishing the scalability and robustness of the amortized zero-shot sampling algorithm.

- While the method performs well with only 1 % of the training data, how does performance evolve as the amount of paired data increases? Is there a clear threshold below which amortization fails to yield reliable speed-ups or reconstruction quality? An analysis of performance versus data scale would strengthen the claim that small subsets suffice.

- What is the computational cost of pre-training the amortized initialization model, and how does this compare to the cost of the zero-shot approach? Is the pre-training complexity included in Table 1 or Figure 1 when comparing total efficiency? A breakdown of training versus inference costs would clarify the overall benefit.

- Could you provide an ablation comparing the trade-off between reconstruction performance and inference time on (i) zero‑shot MGDM, (ii) warm‑start with no subsequent gradient steps, and (iii) warm‑start with varying numbers of steps? This would clarify how much each component contributes.

---

> ### Author Response · Authors · 2025-11-22
>
> Thank you for your review. We added new experiments in **Appendix D** to address all reviewer feedback. We now respond to your comments specifically.
>
> ### Regarding your questions
>
> 1. > Comparison vs. amortized inference with diffusion priors
>
> DAVI [1] is relevant but differs fundamentally. It learns an implicit posterior $q_\phi(\cdot \mid y,\mathcal A)$ from paired data and performs one-step inference, but without test-time likelihood guidance it is tied to the degradations seen during training and fails under shifted operators.
>
> In contrast, our method retains test-time likelihood guidance for OOD robustness while using amortization only to warm-start the variational parameters *within each diffusion step*, yielding both zero-shot robustness and a significant reduction in inference time. To our knowledge, no prior amortized method, including DAVI [1], offers this combination.
>
> 2. > FID evaluation?
>
> With only 300 images, we found FID to be unstable and poorly aligned with perceptual quality. We therefore rely on CMMD [2], which is more sample-efficient and better suited to our experimental setting.
>
> 3. > Gradient step sensitivity?
>
> **Appendix D.5** provides a full ablation for FFHQ-256 motion deblurring and shows that amortized MGDM maintains high reconstruction quality even with very few gradient steps: a warm start with 0 steps already outperforms zero-shot MGDM with 3 steps while being 1.24× faster, indicating that the inference model can largely eliminate the need for gradient updates.
>
> 4. > Select the optimal number of steps?
>
> There is no closed-form rule, because the variational problem in equation (12) is non-convex.
>
> 5. > Higher-resolution datasets?
>
> **Appendix D** now includes new experiments at $256 \times 256$, extending the original $64 \times 64$ results and confirming that the method scales to higher resolution.
>
> 6. > Impact of the amount of paired training data?
>
> **Appendix D.6** now evaluates performance at both 1% and 10% paired data (the submission reported only 1%). Overall, supervised diffusion improves with more data, while amortized MGDM remains robust across data budgets. In particular, it is superior accross all reconstruction metrics compared to supervised diffusion in the regime of 1% paired data.
>
> 7. > Computational cost of training the inference model?
>
> To complement Figure 1 and Table 1, which report only inference time, **Appendix D.4** now compares total training and inference wall-clock costs. For FFHQ-256 motion deblurring (10% training data), training the amortized MGDM model takes under 7 V100-hours. Combined with its $\times 1.4$ speedup over zero-shot MGDM, this implies that amortized MGDM becomes cheaper overall (training + inference) once more than ~2900 images are reconstructed.
>
> ### Regarding the other points you mentionned
>
> 1. > Inaccruate discussion of prior amortized variational inference methods
>
> The final revision will correct the mistake regarding the discussion about [3], as pointed by the reviewer.
>
> However, contrary to the reviewer’s claim, DAVI [1] *does* require paired clean–degraded data for its upstream training, and its OOD robustness results cover only changes in noise level $\sigma_y$. Our evaluation is more challenging: the degradation operator itself changes between training and inference (e.g., different inpainting masks; see **Section 4.3**).
>
> 2. > Supervised diffusion require only a single network pass
>
> No, it require multiple evaluations, one per ODE/SDE timestep.
>
> 3. > The method does not eliminate the test‑time optimization burden
>
> Our method reduces test-time optimization for in-domain operators—**Section 4.1** and **Appendix D.2** show that several gradient steps can be skipped while maintaining quality, yielding a speedup over zero-shot MGDM. For OOD operators, however, achieving both full robustness and faster inference than zero-shot is unrealistic unless the operator remains close to those seen during training.
>
> 4. > Computational cost of the proposed method vs. supervised diffusion
>
> **Appendix D.4** provides a comparison of total computational cost. Although supervised diffusion is faster at inference (5.4 vs. 20.5 s per image), it generalizes poorly to OOD operators (**Section 4.3**) and requires substantially more paired data (**Section 4.2**, **Appendix D.6**). In contrast, our method remains robust under unseen degradations and has a much lower training cost (6.6 vs. 112.5 GPUh).
>
> 5. > Adaptation to other diffusion backbone architecture
>
> The amortization method is architecture-agnostic and applies to standard backbones (UNet, DiT) with minor changes (cf. **Section 3**).
>
> 6. > Dependence on hyperparameter tuning
>
> Hyperparameter tuning is used only to study the quality–speed trade-off, not to favor either zero-shot MGDM or amortized MGDM method.
>
> ### References (cf. submission)
>
> [1] Lee et al., 2024
>
> [2] Jayasumana et al., 2024
>
> [3] Mammadov et al., 2024
>
> [4] Elata et al., 2025

---

### Official Review · Reviewer_CLP9 · 2025-11-01

**Soundness:** 2
**Presentation:** 3
**Contribution:** 2
**Rating:** 2
**Confidence:** 4

**Summary:**

The paper proposes Amortized Variational Diffusion Posterior Sampling (Amortized MGDM), an approach to accelerate variational diffusion posterior sampling by training an inference network that predicts a good initialization for the variational optimization step. This shifts part of the expensive test-time optimization into an offline amortized model trained on a small paired dataset. The method reportedly reduces inference time by up to 32% while maintaining reconstruction quality and improving data efficiency, with preliminary evidence suggesting some robustness to variations in the degradation operator.

**Strengths:**

1. The paper is well-written and well-organized, with detailed reporting of experimental methodology and implementation.
2. The neural network initialization reduces the number of gradient updates required, providing faster inference compared to full iterative optimization.
3. The paper provides a practical engineering contribution that improves the efficiency of an established method during test time.

**Weaknesses:**

1. The reported “up to 32%” inference speedup is measured under specific hyperparameter settings and remains modest considering the additional cost of training the amortized inference network. Moreover, this training step makes the approach no longer zero-shot. Unlike MGDM, which can be applied to arbitrary degradations without retraining, the proposed amortized model must be trained for each degradation operator and dataset. This is a major weakness for the paper.
2. The motivation for the “data-scarce” scenario is unconvincing. In typical image restoration settings, large pretrained diffusion priors (e.g., Palette or Stable Diffusion) already eliminate the need for paired data, making the data-limited justification for amortization weak. If the intended motivation is specialized domains with scarce training data (e.g., medical imaging), this should be demonstrated in such contexts rather than on 64×64 FFHQ. Moreover, in genuinely data-limited applications, a 20–30% slower zero-shot MGDM inference would likely be an acceptable trade-off compared to retraining an amortized model tied to a specific degradation.
3. The paper claims robustness to out-of-distribution (OOD) operators, but both the training and testing setups involve inpainting with different mask patterns. This does not represent a truly OOD scenario, as the degradation type remains the same. Demonstrating genuine robustness would require evaluating across different degradations (e.g., training on inpainting and testing on super-resolution) or different datasets (e.g., training on FFHQ and testing it on ImageNet).
4. The title and framing suggest a conceptual unification of zero-shot and supervised diffusion paradigms, but in reality the method remains a variant of posterior sampling with a learned warm start. Also, since MGDM already includes explicit likelihood guidance, it is expected to be robust to OOD operators, so the novelty is relatively limited.
5. All experiments are restricted to 64×64 FFHQ. There are no results on standard higher-resolution benchmarks (CelebA-HQ 256, FFHQ 256, ImageNet 256, etc.), making it difficult to assess whether the claimed performance generalizes beyond the toy-scale setting.

**Minor Comments:**
- Vectors and matrices should be written in boldface to clearly distinguish them from scalar quantities.
- The choice of measurement noise $\sigma_y$=0.01 is acceptable but notably lower than the conventional setting of $\sigma_y$=0.05. This difference should be acknowledged.

**Questions:**

1. *Regarding weakness 1:* Can the authors report total training and inference wall-clock times (including amortization training) to clarify the end-to-end efficiency trade-off compared to zero-shot MGDM?
2. *Regarding weakness 2:* If the “data-scarce” motivation targets real low-data domains, have the authors considered testing on medical or scientific imaging datasets where this assumption holds? Why was FFHQ chosen instead?

---

> ### Author Response · Authors · 2025-11-22
>
> Thank you for your review. The revised version adds new experiments in **Appendix D** addressing the feedback from all reviewers. We respond to your comments below.
>
> ### Regarding your questions
>
> 1. > Total training and inference wall-clock times?
>
> We now report full training and inference times for zero-shot MGDM, amortized MGDM, and the supervised baseline (**Appendix D.4**, FFHQ-256, 10% data).
> |Method|Train time (GPU h)| Inference time / image (s)|LPIPS|
> |-|-|-|-|
> |Zero-shot MGDM|0|28.7|0.044 ± 0.018|
> |Amortized MGDM (10%)|6.6|20.5|0.045 ± 0.017|
> |Supervised diffusion (10%)|112.5|5.4|0.066 ± 0.020|
>
> Training amortized MGDM is far cheaper than the supervised baseline (6.6 vs. 112.5 GPUh). For $N \in [2898, 25247]$ reconstructions, its overall cost (training + inference) is lower than both zero-shot and supervised baselines.
>
> 2. > Experiments on medical data
>
> This is an interesting direction, but outside our expertise. Our goal in **Section 4.2** is to show that, unlike supervised diffusion, which quickly degrades with few paired samples, the proposed method remains competitive in data-scarce regimes. New 256×256 results in **Appendix D.6** support this.
>
>
> ### Regarding the other points you mentioned
>
> 1. > “Up to 32%” speedup depends on hyperparameters.
>
> Hyperparameter sweeps are used to characterize the quality–speed trade-off for both amortized and zero-shot MGDM. The reported 32% reduction (×1.47 speedup) corresponds to one target LPIPS value, but **Section 4.1** and **Appendix D.2** show consistent speedups across a wide LPIPS range (0.041 to 0.155 on FFHQ-256 motion deblurring), depending on the desired reconstruction quality.
>
> 2. > Must be trained for each degradation.
>
> The approach requires upstream training but is not restricted to a single operator. As explained in **Section 3**, the InvFussion-based inference model conditions explicitly on the degradation operator, so a training set covering multiple operators yields one amortized model that handles all of them. Experiments (**Section 4.1**, **Appendix D.2**) show substantial speedups on all in-domain operators, and **Section 4.3** shows robustness to OOD operators when combined with likelihood guidance.
>
> 3. > Motivation for the data-scarce scenario
>
> The reviewer suggests that large pretrained diffusion priors remove the need for paired data, weakening the motivation for amortization. However, this is not the case: models such as Palette [2] are fully supervised and require clean–degraded pairs; and adapting Stable Diffusion to a specific inverse problem still requires paired data to learn the corresponding conditional mapping.
>
> Our claim concerns the regime where paired data are limited. In this setting, the proposed method remains competitive, whereas supervised diffusion degrades (**Section 4.2**, **Appendix D.6**).
>
> 4. > OOD accross different degradations and datasets
>
> **Appendix D.7** adds cross-operator OOD experiments: models trained only on motion deblurring and tested on ×4 super-resolution. The supervised baseline (Palette) collapses, while amortized MGDM maintains strong reconstruction quality. Cross-dataset evaluation is orthogonal to our focus on operator-shift robustness and left for future work.
>
> 5. > Expected OOD robustness reduces novelty
>
> Because the inference model is trained only on *in-domain* operators, warm-starting variational inference on *OOD operators* could in principle lead to suboptimal optimization in the variational problem and worse results than zero-shot MGDM. It is therefore important to verify that amortization does not harm OOD performance. As shown in **Section 4.3**, amortized MGDM matches zero-shot MGDM on OOD operators, despite being trained solely on in-domain degradations, while supervised diffusion models degrade severely.
>
> 6. > All experiments are restricted to 64×64
>
> The revision includes 256×256 experiments in **Appendix D**, showing that the method scales beyond 64×64. We initially used 64×64 for a strictly controlled comparison across the methods.
>
> 7. > Not a conceptual unification
>
> In this work, our goal is to highlight the gap between supervised and zero-shot diffusion: supervised methods rely on upstream training on paired data but do not perform test-time likelihood optimization, making them brittle under OOD degradations; zero-shot methods like MGDM rely solely on test-time optimization and do not benefit from upstream training.
>
> Our method offers one concrete way to combine these two paradigms: it uses both an upstream training phase and test-time computation. This yields (i) faster inference than zero-shot MGDM, (ii) competitive reconstruction quality even with limited paired data (**Section 4.2** and new **Appendix D.6**), and (iii) improved robustness to OOD operators (**Section 4.3**).
>
> 8. > Minor comments
>
> We will incorporate these in the final revision.
>
> ### References (cf. submission)
>
> [1] Elata et al., 2025
>
> [2] Saharia et al., 2022

---

### Official Review · Reviewer_bgvG · 2025-11-03

**Soundness:** 2
**Presentation:** 2
**Contribution:** 2
**Rating:** 4
**Confidence:** 3

**Summary:**

The work concerns inverse problems such as inpainting, where a model is trained over paired data to warm-start the inference time variational approximation as opposed to zero-shot variational posterior sampling and supervised diffusion for inference. The authors claim up to 32% inference time savings and data scarce training possibilities, in addition to robustness to OOD operators.

**Strengths:**

1. The work evaluates comprehensively using PSNR/SSIM. It also reports LPIPS, CMMD, shows qualitative samples, and evaluates (i) speed/quality tradeoff, (ii) low-data regime, and (iii) OOD degradations (new masks), covering decent breadth of evaluations.
2. The method is a simple addendum to existing approaches and demonstrates promising results under the chosen evaluation regime.

**Weaknesses:**

1. OOD robustness only works upon adding back the additional gradient steps, thus falling back on the behavior on MGDM.
2. Data scarce regime comparisons are only made to Palette which may not be suited for the 1% setting. Additional methods that target scarcity in data could be incorporated to enhance one of the work's core claim of data scarcity robustness. The bulk of the benefits appear to arise from the pre-training in the existing conditional denoisers and the MGDM's framework, which Palette doesn't have.
3. All experiments are limited to 64x64 image sizes in facial domain.
4. The authors could provide some insights into why the proposed amortization must preserve correctness and its limitations.

**Questions:**

1. Could the authors provide insights into which steps lead to the time saving that is being observed?
2. How far is the amortized network's prediction from the optimal setup with full optimization?
3. Does the method generalize to larger images where MGDM struggles?

---

> ### Author Response · Authors · 2025-11-21
>
> Thank you for your review. In the revised version, we added new experiments in **Appendix D** to address the feedbacks provided by all reviewers. We now respond to your comments specifically.
>
> ### Regarding your questions
> 1. > Insights into which steps lead to the time saving
>
> The speed-up of amortized MGDM comes from reducing the number of gradient steps needed for variational approximation, thanks to the inference network that provides a warm start instead of a zero-shot initialization.
>
> **Appendix D.5** includes a new ablation on FFHQ-256 motion deblurring to give more insights on this. We give a summary here:
> |Method|# Gradient Steps|LPIPS|Inference Time|Notes|
> |-|-|-|-|-|
> |Zero-shot MGDM|3|0.085 $\pm$ 0.034|17.8 s|Requires iterative optimization|
> |Amortized MGDM|0|0.072 $\pm$ 0.027|14.3 s|×1.24 faster|
>
> This shows that the inference network can replace most (or all) gradient steps while preserving reconstruction quality.
>
> 2. > How far is the amortized network's prediction from the optimal setup with full optimization?
>
> The variational approximation problem in equation (12) of the submission is non-convex because it involves the unconditional denoiser parameterized as a neural network. Consequently, the optimal solution to this optimization problem is unknown.
>
> To evaluate how well the amortized inference network $c \mapsto (\mu_\varphi(c), \rho_\varphi(c))$ approximates the solution of the variational objective, we can still compare its loss   $L_1(c) = \mathcal{L} (\mu_\varphi(c), \rho_\varphi(c); c)$ with $L_2(c) = \mathcal{L}(\mu_{\mathrm{zs}}, \rho_{\mathrm{zs}}; c)$,  where $(\mu_{\mathrm{zs}}, \rho_{\mathrm{zs}})$ are obtained by initializing from the zero-shot prior and running $G = 10$ Adam steps (with learning rate $\eta = 0.03$). We perform here this comparison on an FFHQ-256 motion deblurring example (image index 61000), where we consider all contexts $c$ involving timesteps $t \in \mathcal{T}$ during MGDM inference, where $\mathcal{T}$ is the subset of timesteps on which the inference model is trained.
>
> |Metric|Value|
> |-|-|
> |LPIPS|0.047|
> |SSIM|0.915|
> |PSNR|31.2|
> |Min $L_1(c) / L_2(c)$|0.506|
> |Median $L_1(c) / L_2(c)$|0.930|
> |Max $L_1(c) / L_2(c)$|0.958|
>
> The amortized model consistently obtains losses lower than those achieved after 10 gradient updates from the zero-shot initialization, indicating that it provides sufficiently accurate variational parameters without  iterative optimization.
>
> 3. > Generalization to larger images
>
> The **Appendix D** in the revision now contains new experiments demonstrating that the method remains effective on larger $256 \times 256$ images, beyond the original $64 \times 64$ resolution.
>
> Note that our initial choice of 64×64 FFHQ reflected the need to ensure a *controlled* comparison where all methods (zero-shot MGDM, amortized MGDM, supervised diffusion) are trained and evaluated under matched configurations.
>
>
> ### Regarding the other points you mentioned
>
> 1. > OOD robustness only works after adding gradient steps.
>
> Our goal is not to surpass zero-shot MGDM on OOD operators—which is generally unrealistic without overlap between in-distribution and OOD degradations—but to verify that amortization does not degrade OOD performance. Experiments show that amortized MGDM remains competitive: although the warm start provided by an inference model trained only on in-domain operators is weaker on OOD degradations, a few gradient steps reliably recover the same quality as zero-shot MGDM, as shown in **Section 4.3**. In contrast, supervised diffusion performs poorly on OOD operators.
>
> 2. > Additional methods that target scarcity in data could be incorporated
>
> The reviewer did not precise which methods are referred to here. Our baselines, Palette [1] and InvFussion [2], are standard and competitive. The conditional denoiser in these supervised baselines is initialized from a pretrained unconditional denoiser and fine-tuned on 1% paired data, with hyperparameter tuning to ensure strong performance in the low-data regime, cf. **Section C and D.1**.
>
> 3. > Insights on the limitations of the method
>
> A limitation is that reconstruction quality on in-distribution operators does not surpass a well-trained supervised baseline when large paired datasets are available. However, amortized MGDM is more effective in low-data regimes (**Section 4.2**). Note that the fact amortized MGDM is not doing a better reconstruction quality than zero-shot MGDM on OOD operators is *not* a limitation of the method, as explained above.
>
> Future work could explore how to better leverage large-scale paired datasets during the upstream training phase to match the supervised baseline on in-domain operators, *while maintaining superior robustness* to out-of-distribution operators, as demonstrated in this work.
>
> ### References
>
> [1] Palette: Image-to-image diffusion models, 2022
>
> [2] InvFussion: Bridging Supervised and Zero-shot Diffusion for Inverse Problems, 2025

---

### Author Response · Authors · 2025-12-03
**Discussion summary and final revision**

We thank the reviewers once again for their valuable feedback. We have updated our final revision accordingly, with all changes relative to the initial submission marked in magenta. Key changes include:
* Experiments on ImageNet in **Appendix E** showing a speedup of amortized MGDM over zero-shot MGDM (**this is new since our last rebuttal**).
* Extension of the experiments in the larger resolution $256 \times 256$ in **Appendix D**, as requested by the reviewers.
* Several new ablation studies in **Appendix D**, as requested by the reviewers.
* Clarification of misunderstandings and concerns raised in the reviews (see below).


Below, we summarize the key concerns of the reviewers throughout the discussion period, and how our rebuttal addresses these concerns.

## Concern 1: Clarification on OOD robustness
*Reviewers expected amortized MGDM to outperform zero-shot MGDM under OOD degradations.*

**Our rebuttal**: This is not the objective. Outperforming zero-shot MGDM on unseen degradations is unrealistic without overlap between training and OOD operators. Our aim is to demonstrate robustness vs. supervised diffusion and ensure that amortization (trained only on in-domain operators) does not harm OOD performance of MGDM. Our experiments in **Section 4.3** show that, even when the inference model is evaluated on unseen operators, adding gradient steps recover zero-shot quality. We further strengthened the evaluation with a cross-operator OOD experiment (training on motion deblurring, testing on ×4 super-resolution): supervised diffusion fails on OOD settings, while AMGDM remains robust due to its test-time likelihood guidance.

## Concern 2: Scalability beyond 64×64 and face images
*Reviewers were concerned that the method might not scale beyond face images (FFHQ) and the resolution $64 \times 64$.*

**Our rebuttal**: New 256×256 experiments in **Appendix D** confirm that *all claims*—speedup, quality, robustness to OOD and data-scarcity—hold at higher resolution. Amortized MGDM consistently accelerates MGDM at matched quality, while supervised diffusion remains fast but fails in OOD and low-data regimes. **Appendix E** further confirms on natural images (ImageNet instead of FFHQ) that the acceleration claim for amortized MGDM still holds.

## Concern 3: Computational cost and usefulness of amortization
*Reviewers questioned whether amortization yields meaningful gains.*

**Our rebuttal**: Training amortized MGDM is *light* (6.6 h on a single NVIDIA V100 GPU) compared to supervised diffusion (112.5 GPUh). At inference, amortized MGDM provides ~1.4× speedup vs. zero-shot MGDM, and when reconstructing ≥ 2900 images, its total cost (training + inference) becomes *lower* than zero-shot MGDM. Importantly, training an inference model is not tied to a single degradation operator, because its architecture is based on InvFussion [1] (see **Section 3**), meaning that the inference model can handle several degradation operators.

## Concern 4: Comparison to related amortized methods
*Reviewers found novelty unclear compared to existing amortization methods (e.g., DAVI [2]).*

**Our rebuttal**: The novelty of our method is to perform amortization *within* each diffusion step, in contrast to DAVI, which performs a *single-step* amortization without test-time likelihood guidance. To our knowledge, our approach is the *only* approach combining upstream training with test-time likelihood optimization, yielding (i) acceleration relative to zero-shot MGDM and (ii) OOD robustness absent in supervised diffusion and prior amortized methods.

---

### Meta-Review · Area_Chair_Razt · 2026-01-06

**Summary:**

The paper introduces an amortized initialization for MGDM-based diffusion posterior sampling. The core idea is training an inference network to warm-start the variational optimization, aiming to cut down test-time costs. Reviewers broadly accepted the motivation. However, the main concerns were the incremental nature of the novelty (essentially learning a good initializer for an existing sampler), the practical trade-off between the added training overhead and the resulting speedup, and whether the method scales beyond the initial toy settings, i.e., limited breadth of evidence across tasks/operators/baselines.

**Reviewer Concerns:**

Concerns addressed:
1) The rebuttal adds evidence beyond the original setting (including higher-resolution / additional datasets) and clarifies generality claims.
2) It provides clearer accounting of training vs inference cost, helping interpret when amortization is worthwhile.
3) It improves context vs related posterior sampling baselines.

Concerns remained:
1) Novelty/impact remains limited since the main contribution is amortization of an existing optimization-based sampler.
2) The practical payoff still appears moderate relative to the additional training requirement and paired-data assumption.
3) Evidence is improved but still not broad enough to demonstrate consistently strong end-to-end gains across a wide range of operators/problems and strong baselines.

**Reviewer Scores:**

1) Reviewer (score 4): likely unchanged at 4; the rebuttal strengthens evidence but they were already borderline-positive.
2) Reviewer (score 4): likely unchanged at 4 (possibly slightly more positive, but still borderline).
3) Reviewer (score 2): likely increase to 3 given added experiments and clearer cost breakdown, but unlikely to flip due to novelty/impact concerns.
4) Reviewer (score 2): likely unchanged.

---

### Decision · Program_Chairs · 2026-01-26

Reject